# Stable choice coding in rat frontal orienting fields across model-predicted changes of mind

J. Tyler Boyd-Meredith [1,4], Alex T. Piet [1,2,4], Emily Jane Dennis [1], Ahmed El Hady [1,5✉] & Carlos D. Brody [1,3,5✉]

During decision making in a changing environment, evidence that may guide the decision accumulates until the point of action. In the rat, provisional choice is thought to be represented in frontal orienting fields (FOF), but this has only been tested in static environments where provisional and final decisions are not easily dissociated. Here, we characterize the representation of accumulated evidence in the FOF of rats performing a recently developed dynamic evidence accumulation task, which induces changes in the provisional decision, referred to as "changes of mind". We find that FOF encodes evidence throughout decision formation with a temporal gain modulation that rises until the period when the animal may need to act. Furthermore, reversals in FOF firing rates can be accounted for by changes of mind predicted using a model of the decision process fit only to behavioral data. Our results suggest that the FOF represents provisional decisions even in dynamic, uncertain environments, allowing for rapid motor execution when it is time to act.

[1] Princeton Neuroscience Institute, Princeton University, Princeton, NJ, USA. [2] Allen Institute, Seattle, WA, USA. [3] Howard Hughes Medical Institute, Princeton University, Princeton, NJ, USA. [4] These authors contributed equally: J. Tyler Boyd-Meredith, Alex T. Piet. [5] These authors jointly supervised this work: Ahmed El Hady, Carlos D. Brody. ✉email: ahady@princeton.edu; brody@princeton.edu

When making decisions, animals must weigh and combine the available evidence in favor of each alternative. With each new observation, evidence about the underlying state of the environment gradually accumulates until the animal is ready to act. This accumulation model successfully describes a wide array of decisions[1–3]. Neural correlates of this accumulation process are also present across many brain regions in animals performing perceptual categorization tasks[1,4]. Not all brain regions with neural correlates of evidence accumulation play the same role in the decision making process[4–6]. For example, regions important for accumulation may represent evidence in a continuous, graded fashion. On the other hand, regions important for reading out choice and preparing motor movements may have more categorical representations of the accumulated evidence.

Hanks et al.[7] characterized the neural representation of accumulating evidence in rats performing accumulation of trains of auditory click evidence. In the task, two streams of randomly-timed auditory clicks were emitted from either side of a fixation location and rats were trained to orient toward the side that played a greater number of clicks. Presenting the evidence as discrete pulses provided additional power to estimate the evolution of each subject's latent accumulated evidence variable on individual trials[8], increasing the resolution for estimating neural encoding of this variable across brain regions[7,9,10]. Experimenters recorded from the posterior parietal cortex (PPC) and the frontal orienting fields (FOF), a frontal cortical structure implicated in short term memory and preparation of orienting movements[11–13]. They found that FOF neurons encoded the instantaneous accumulated evidence with sigmoidal tuning curves that remained stable during accumulation[7]. These representations were more categorical than representations found in PPC, providing a readout of the animal's provisional decision—the choice favored by the evidence presented so far—throughout accumulation[7,10]. While this study could not differentiate between evidence representations resulting from a role in motor preparation and motor-independent evidence representations, temporally-precise perturbations of the signals in FOF only impaired the animal's choice when they overlapped with the final time points of accumulation and not when they occurred early in the evidence period. These results, along with a two-node model of the FOF[14], suggested that the FOF is not involved in the accumulation of new pieces of evidence, but provides a critical readout of the animal's provisional decision when it is time to act.

While these experiments were conducted using stationary environments, many natural decisions unfold in dynamic environments. In stationary settings, all evidence samples in a trial reflect the same underlying environmental state. This means the best strategy is to equally weigh all samples of evidence throughout stimulus presentation[15]. In this regime it is difficult to dissociate the provisional from the final decision. In dynamic environments, the state of the world can change while the animal is deliberating. This means the animal should learn to discount old evidence via leaky integration, weighing more recently presented samples of information more heavily than older samples[16–20]. Unlike stationary environments, adopting the optimal strategy in a dynamic environment leads to frequent fluctuations in the animal's provisional decision.

Recent work has shown that rats and humans can learn to adopt the optimal discounting rate in a dynamic environment[16,18]. However, it is unknown whether the neural correlates of evidence accumulation observed during putatively non-leaky integration in stationary environments are preserved in animals performing putatively leaky integration in dynamic environments. Here, we recorded from FOF in rats during a dynamic accumulation of evidence task. We tested whether the stable code observed in the stationary environment persisted in the dynamic environment by applying and extending a method developed to characterize neural tuning to accumulated evidence[7]. The evolution of the latent accumulation variable was estimated using a behavioral model fit to the animal's choice data[18]. In FOF, tuning to this accumulation variable was described by a single sigmoidal tuning curve multiplied by a time varying gain modulation, which increased with time early in the trial and stabilized at the time of the earliest possible go cue.

We reasoned that if FOF neurons track the accumulated evidence throughout the entire accumulation period, firing rates should respond rapidly to changes in the provisional decision, which in the literature are referred to for short as "changes of mind". Such "changes of mind" have been studied in stationary environments when movement trajectories initiated toward one target are subsequently revised, possibly due to continued processing of the stimulus after initial decision commitment[21,22]. They may also arise from noisy fluctuations in decision-related neural activity[23] and their timing may be inferred through neural decoding[24,25]. (For clarity, we emphasize that we do not claim to test whether the FOF encodes an abstract notion of "mind", but much more simply that changes in the provisional decision can be read out from FOF activity). We used a behavioral approach to predict the precise timing of changes of mind using the latent state of the behavioral model fit to each rat's choice data. We found that FOF neurons responded to these model state change events within 100 ms, reflecting the new provisional decision in their activity. Recomputing the evidence tuning curves aligned to model state changes, we confirmed that FOF neurons encode evidence with a single tuning curve before and after changes of mind. These results suggest that FOF maintains a stable readout of the decision provisionally favored by the accumulated evidence despite dynamic uncertainty in the environment and the upcoming choice. Maintaining a stable representation of the provisional decision may help ensure that the animal is ready when it is time to act. Our study opens up the opportunity for future work on the neural circuit level understanding of how animals integrate and decide in a volatile environment.

## Results

**The dynamic evidence accumulation task**. We trained rats ($n = 5$) to perform a previously developed dynamic evidence accumulation task[18]. This task requires the rat to report which of two hidden states the environment is in at the time of a go cue. At the beginning of each trial, the center port in an array of three nose ports is illuminated by an LED. This invites the rat to poke its nose into the center port, initiating presentation of an auditory stimulus. The stimulus is composed of two trains of auditory pulses (clicks) delivered in stereo from speakers positioned on either side of the center port. The left and right click trains are generated from different Poisson processes with rate parameters, $r_R^i$ and $r_L^i$, that depend on the state $i$. When the environment is in state 1, the "go right" state, the generative click rate is higher for the right speaker than the left ($r_R^1 = 38 Hz$ and $r_L^1 = 2 Hz$). In state 2, the "go left" state, the click rates are reversed ($r_R^2 = 2 Hz$ and $r_L^2 = 38 Hz$). Trials begin in either state with equal probability and switch stochastically between states with a fixed hazard rate $h = 1 Hz$. After a randomized duration, drawn from a uniform distribution between 500 and 2000 ms, the stimulus ends and the center LED turns off. This "go" cue signals the rat to withdraw from the center port and poke its nose into one of two reward delivery ports on either side. The animal receives a drop of water (18 uL) if it chooses the side port corresponding to the final value of the hidden state. Incorrect choices were signaled with a white noise stimulus (Fig. 1a). In our dataset, roughly 33% of trials had

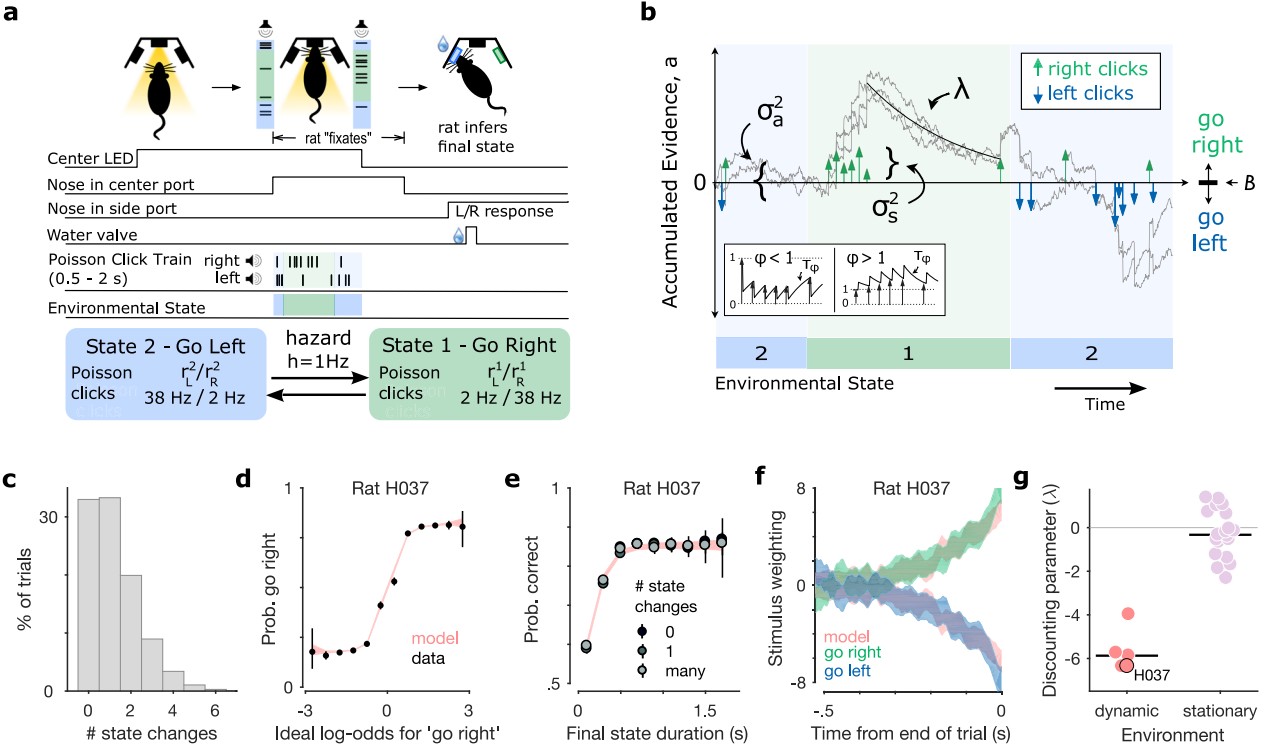

**Fig. 1 Rats accumulate and discount evidence in a dynamic accumulation task. a** Schematic showing task events and timing. The center port is illuminated by an LED. The rat pokes its nose into the port to initiate playback of randomly timed auditory clicks from speakers on either side. Clicks on each side are generated with different underlying Poisson rate parameters that depend on a hidden environmental state. The stimulus duration is drawn from a uniform distribution between 500 and 2000 ms. During that time the hidden state changes stochastically at a fixed hazard rate, $h = 1 Hz$. At the end of the stimulus presentation, the center LED turns off and reward is baited in the side port corresponding to the final state. **b** Schematic of the evolution of the accumulation model on an example trial. Three example accumulation traces are shown for different instantiations of the noise applied at each time point ($\sigma_a$) and the noise applied to each click ($\sigma_s$). Neighboring clicks can either depress or facilitate each other according to the adaptation parameters ($\phi$ and $\tau_\phi$). The evidence discounting rate ($\lambda$) determines how quickly the decision variable $a$ decays back to zero. At the end of the trial, a choice is made by comparing the decision variable to the decision boundary parameter $B$. **c** Frequency of state changes per trial across all rats' datasets. **d** Example psychometric plot showing the probability that the rat chooses "go right" as a function of the ideal observer log-odds supporting a "go right" choice. Rat data (black points) is overlaid on predictions of the accumulation model with parameters fit to this rat (red traces). Errorbars for rat data represent 95% binomial confidence intervals around the mean ($n = 92,468$ trials from 252 sessions). **e** Example final state chronometric plot for the same dataset as in (**d**). Accuracy (mean with 95% binomial confidence intervals) is plotted as a function of the duration of a trial's final state and the number of state changes in a given trial. **f** Psychophysical reverse correlation kernel for the same dataset as in (**d**) and (**e**). Green and blue patches indicate strength (mean ± s.d.) of evidence favoring rightward choice as a function of time until the trial ends for rightward and leftward choices, respectively. The red patches are corresponding predictions from the accumulation model. **g** Discounting parameters for each rat in this study (red points) compared to each rat in a previously published stationary environment (lilac points; Brunton et al.[8]). Group medians are plotted as black horizontal lines. Source data are provided as a Source Data file.

no state changes, 33% had one, and 34% had more than one (Fig. 1c).

**Behavioral model captures leaky integration strategy**. We fit a previously-developed behavioral model[8,18] to rats' choices using an average of 108,126 trials per rat (63,494 to 185,091 trials each from 118 to 308 sessions). The model (Fig. 1b) parameterizes the process by which the evidence available in each auditory click is integrated over time into a decision variable that guides the rat's choice.

The decision variable, referred to as the accumulation value $a$, takes an initial value $a_0$, drawn from a Gaussian with zero mean and an initial variance $\sigma_i^2$, which is fixed across trials. Each right and left click increments or decrements the accumulation value, subject to sensory adaptation governed by parameters $\phi$ and $\tau_\phi$. Each click also introduces additional noise with variance $\sigma_s^2$. Memory noise with variance $\sigma_a^2$ is introduced at each time step. Evidence is discounted with rate $\lambda$, which parameterizes the rate at which, in the absence of further input, $a$ decays with time

($\lambda < 0$) or increases with time ($\lambda > 0$). When $\lambda < 0$, older pieces of evidence are discounted relative to newer evidence. While decision makers in stationary environments perform best when discounting is minimal ($\lambda = 0$), ideal observers in our task adopt a high-level of discounting of old evidence ($\lambda < 0$), reducing the impact of older clicks that may have been presented before a change in the hidden state[16–19]. As previously described[18], the optimal discounting rate in a dynamic environment depends on the quality of evidence, including the observer's per-click noise $\sigma_s^2$. At the end of non-lapse trials, the rat chooses to go right if the final accumulation value $a_N$ is greater than the decision boundary $B$, and chooses to go left if $a_N < B$. The ideal value of $B$ is 0 and any deviation from 0 reflects the animal's side bias. On a fraction of trials $l$, called "lapse" trials, the rat chooses randomly. Unlike the model described by Brunton et al.[8], there is no decision bound setting the maximum magnitude of $a$ at which the animal is fully committed to a decision. Instead, the decision remains sensitive to new information throughout the accumulation period. Best fit values of this parameter were previously found to be effectively

infinite and did not improve model fits[8]. For each rat, we used maximum likelihood estimation to find the parameter set $\theta$ that best described the rat's choices across all behavioral trials (see methods for mathematical details; see Supplementary Fig. 1 for all rats' best fit parameters). This model is highly flexible and can capture many possible behavioral strategies[8,18].

We present several assessments of task performance and model validation. Psychometric curves show a rat's choices as a function of the ideal observer log-odds favoring a rightward choice, as well as the correspondence with predictions from the behavioral model fit to an example rat (Fig. 1d) and all rats used in this study (Supplementary Fig. 2). Final state chronometric curves show that performance increased with the final state duration, the elapsed time between the final state change and the "go" cue (Fig. 1e and Supplementary Fig. 3). Radillo et al.[19] demonstrated the rate of increase and saturation level of the chronometric curve for an ideal observer depends only on the hazard rate and SNR of the click rates. Psychophysical reverse correlation kernels quantify the influence of clicks at each timepoint throughout the stimulus period, providing an assay of the rats' evidence discounting. Reverse correlations for all rats in this study show heavier weighting of clicks presented at the end of the trial compared to the beginning (Fig. 1f and Supplementary Fig. 4).

The behavioral model parameter fits for each rat confirm that all rats used a leaky integration strategy ($\lambda < 0$). Best fit discounting parameters were significantly different from a previously reported[8] dataset of rats integrating in a stationary environment ($p < 0.01$; two-tailed Wilcoxon rank-sum test, $n = 5$ in dynamic and $n = 19$ stationary environments) (Fig. 1g). Consistent with previous work, rats adopted discounting rates that favor more recent evidence due to the environmental volatility[18]. The best fit model parameters, along with the model-independent behavioral assays described above, provide converging lines of evidence that the rats integrated evidence throughout the trial, with hundreds of milliseconds and multiple sensory clicks influencing their final decision.

**FOF responses during dynamic accumulation**. We recorded from the frontal orienting fields (FOF) of rats performing the dynamic evidence accumulation task. In 4 rats, we implanted unilateral ($n = 2$ left FOF, 2 right FOF) microwire arrays at coordinates ($+2$ AP; $\pm 1.3$ ML) (Fig. 2a). In a 5th rat, we implanted a bilateral tetrode drive over the same coordinates. Recordings from 69 sessions yielded 738 units across 5 animals. See Supplementary Table 1 for a breakdown of data by rat (Method, location). Cells were considered active and included for further analysis if they had a mean firing rate of at least 1 Hz during the trial ($n = 579$ active cells).

Individual cells show stereotyped temporal dynamics aligned to both the onset of the trial (entering the center nose port), and the movement following the end of the stimulus (nose out of center port). Many individual cells had trial-averaged firing rates that diverged throughout the trial, reaching a final value corresponding to the animal's choice (Fig. 2b; see Supplementary Fig. 5 for spike rasters). These cells had more intermediate average values throughout error trials, but eventually diverge according to the animal's choice, suggesting that firing rates reflect the animal's internal representation of the evidence or the motor plan. We tested the timecourse of selectivity for single neurons to right versus left choices by computing the area under the receiver operating characteristic curve (AUC) and comparing it to a permutation distribution computed by shuffling choice labels across trials. For purposes of visualization, cells are sorted by latency to 200ms (8 consecutive time bins) of significant AUC values (2-tailed permutation test, 250 permutations, $p < 0.05$). We

present these plots for all active neurons and for a subset of pre-movement side-selective neurons (Fig. 2c). Cells were defined as pre-movement side-selective if their total spike counts during the trial between the start of the stimulus and the movement away from the fixation port were significantly different depending on the animal's side choice (2-tailed $t$ test, $p < 0.05$). This subset made up 17.8% of the active population ($n = 103$ selective). For each neuron, the side associated with the higher spike count is referred to as the cell's preferred side. Following Hanks et al.[7], we focus on these pre-movement side-selective neurons because they are most likely to play a role in decision formation.

Pre-movement side selectivity was slightly less common in this dataset than in previous studies of FOF in stationary environments[7]. This may be a consequence of frequent changes of mind, which create a dissociation between provisional and final choice throughout trials in the dynamic task. Across pre-movement side-selective neurons, we computed the average activity conditioned on final state duration and cell preference (Fig. 2d). We observe divergences at different latencies depending on the final state duration (see Fig. S6 for population average conditioned on side-choice and trial outcome as in Fig. 2b).

**Stable accumulator tuning in dynamic environment**. The choice-selectivity metrics presented above reveal coding of the final choice in average neural activity. However, during the trial, the hidden state can change multiple times ($1.22 \pm 1.20$ state changes per trial). This creates frequent dissociations during the trial between the animal's provisional choice and the final choice, which are rare in stationary environments. To better describe encoding of the provisional choice throughout the trial, we applied and extended a method developed to quantify the tuning of single neurons to the accumulation value at each moment during the trial. Grouping firing rates according to the predicted accumulation values at each timepoint, allows us to more informatively combine information across trials with different hidden state change timing, final choice, and trial outcome. Using this method, Hanks et al.[7] found that FOF neurons had a stable encoding of evidence throughout accumulation in a stationary environment. Here, we sought to test whether the FOF continues to stably encode the evidence throughout trials when the environment is dynamic. We reasoned that choice-encoding might emerge later in the dynamic environment when early provisional decisions are less likely to be acted upon. For example, neurons might only represent choice once the stimulus period has ended and the animal has committed to a decision. Further, we asked whether this encoding can still be captured by a single tuning curve in the dynamic environment. While our cell selection enforces that there be choice-encoding at some point in the trial, it does not constrain the presence or stability of this encoding over the course of the trial.

We used the approach described by Hanks et al.[7], to produce a map describing each neuron's tuning to the accumulation value over the course of the trial. First, we computed the joint distribution $P(r, a, t)$ of each cell's firing rate $r$, the instantaneous accumulation value $a$, and time in the trial $t$. The evolution of the distribution over $a$ in response to right and left click trains $\delta_R$ and $\delta_L$, given by the behavioral model described above, was further constrained using the animal's choice $y$ on each trial, giving the posterior distribution

$$p(a) = P(a|t, \delta_R, \delta_L, \theta, a_0 \sim \mathcal{N}(0, \sigma_i^2), y). \tag{1}$$

We improve on the method used by Hanks et al.[7] by using an analytical computation of the posterior distribution of accumulated evidence, allowing for more accurate estimation of $P(r, a, t)$ (see methods). Firing rate maps are generated by computing the

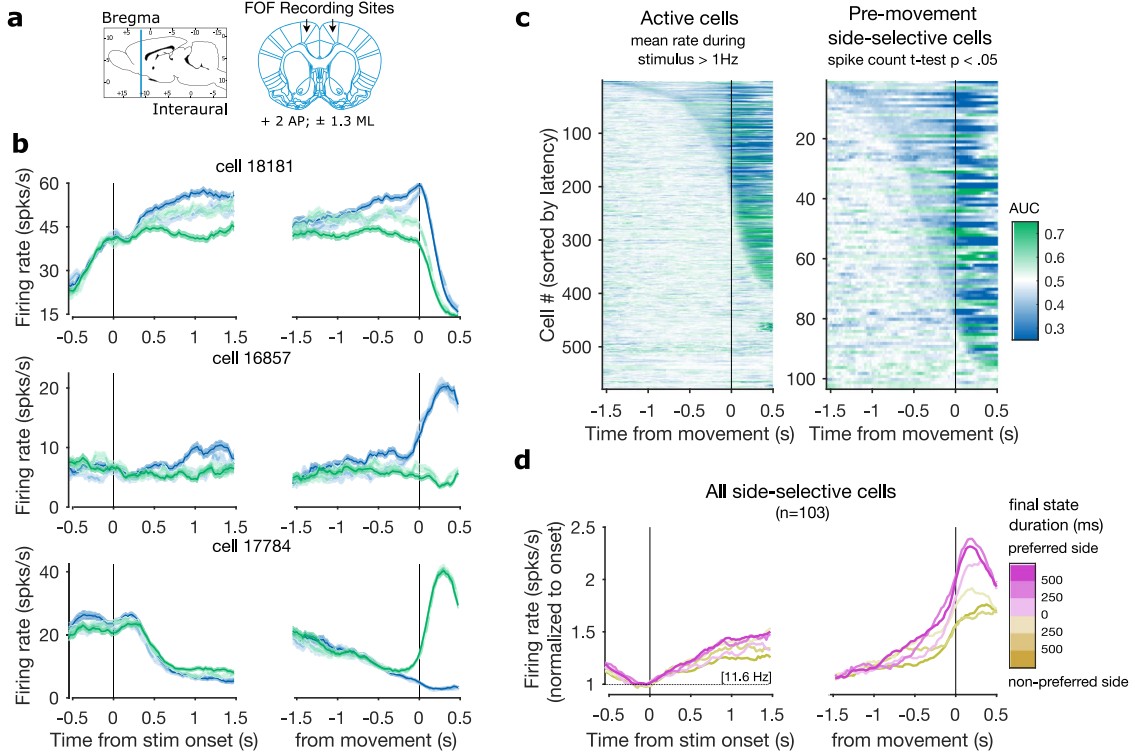

**Fig. 2 FOF neurons encode the rat's upcoming choice. a** Coordinates used for FOF recordings (+2 AP; ± 1.3 ML). **b** Average firing rates for three example FOF cells aligned to stimulus onset (left) and movement (right). Activity is conditioned on right (green) vs. left (blue) side choice, as well as hits (solid lines) vs. errors (dashed lines). Shaded regions represent s.e.m. **c** Side-selectivity at each time point relative to movement for all active cells (left; firing rate > 1 Hz) and for the subset of these cells that meet the spike count pre-movement side-selectivity criterion (right; 2-tailed *t* test *p* < 0.05). AUC is computed on spike rates for right versus left choices. Plots are sorted by latency to 200 ms (8 consecutive time bins) of significant AUC values relative to a distribution created by permuting choice labels across trials (2-tailed permutation test, 250 permutations, *p* < 0.05). **d** Average activity of all pre-movement side-selective cells conditioned on final state duration and cells' side preferences. Grand-average firing rate at stimulus onset (11.6 Hz) is written in brackets. Source data are provided as a Source Data file.

conditional expectation of the firing rate as a function of accumulated evidence and time, for each cell $E[r|a, t]$. We present this rate map for an example cell which is strongly tuned to the accumulator throughout the trial, firing more when accumulated evidence favors left choices (Fig. 3a). Because our neurons have stereotyped temporal dynamics aligned to stimulus onset, we subtract out the average temporal dynamics to isolate $E[\Delta r|a, t]$ (Fig. 3b), the expected firing rate modulation by accumulated evidence over time

$$E[\Delta r|a, t] = E[r|a, t] - E[r|t]. \tag{2}$$

Following Hanks et al.[7], a summary tuning curve was computed by averaging over time to get $E[\Delta r|a]$ (Fig. 3c).

We extend the method by computing the rank 1 approximation of the residual firing rate map $E[\Delta r|a, t]$ using the singular value decomposition (Fig. 3d). For the example cell, this approximation captures 99.6% of the variance in the estimated residual firing rate map. The mean variance explained by this approximation for all pre-movement side-selective cells was 89.7% ± 9.8% (Fig. S10). Higher explained variance indicates that a cell's residual rate map can be accurately described by a single tuning curve with linear scaling across time points. The fraction of the variance captured by the rank 1 decomposition is positively correlated with the total duration of side-selectivity favoring the cell's preferred side (Pearson's correlation, $\rho = 0.41$, $p < 0.01$; Fig. S10C). The approximation is equal to the outer product of the first left singular vector $u_1$ and the first right singular vector $v_1$, scaled by the first singular value $s_1$. These terms can be rearranged and interpreted as the outer product of a firing rate modulation,

$\hat{m}(t) = u_1 s_1 \operatorname{range}(v_1)$ and a tuning curve $\hat{f}(a) = v_1 / \operatorname{range}(v_1)$. Scaling by range($v_1$) gives $\hat{f}(a)$ unit scale and $\hat{m}(t)$ units of spikes/s. Our complete tuning curve approximation becomes:

$$r(a, t) \approx E[r|t] + \hat{m}(t) * \hat{f}(a). \tag{3}$$

We computed a population average residual rate map across all pre-movement side-selective cells by computing the residual firing rate map $E[\Delta r|a, t]$ for each cell using *z* scored firing rates. The accumulated value axis was inverted for left choice preferring cells and then the residual firing rate maps were averaged together (Fig. 3e). We computed the rank 1 approximation of this population residual rate map. This approximation explained 99.7% of the variance in the population residual rate map (Fig. 3e middle, right). The population firing rate modulation curve $\hat{m}(t)$ rises for the first 500 milliseconds and then plateaus at its maximum value. Therefore, the population tuning can be described as a single tuning curve whose modulation increases during the period of the trial before a "go" cue is possible. The modulation stabilizes at its maximum value during the period in which the trial may end and the animal may need to report its decision. Despite the dynamic environment, and changing provisional choice, we find FOF neurons continue to stably encode the evidence with a single tuning curve throughout evidence accumulation.

**Neurons track model-predicted changes in provisional decision.** If cells are stably tuned to the accumulated evidence throughout deliberation, we should be able to see rapid responses in their firing rates to changes in the animal's provisional

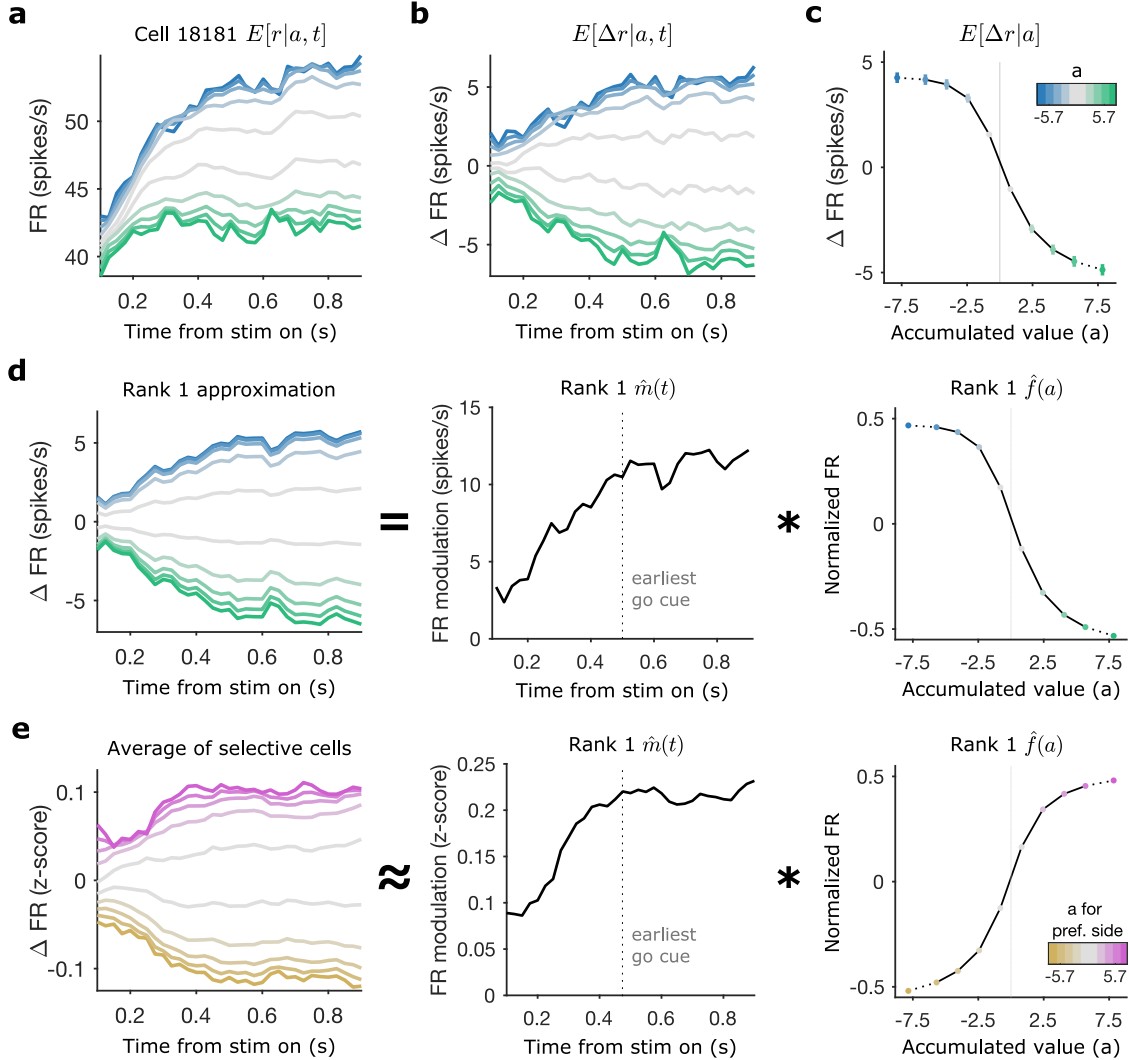

**Fig. 3 FOF neurons encode the accumulated evidence throughout the trial despite a changing environment. a** Firing rate map as a function of accumulated evidence and time for an example neuron. Colors indicate accumulated evidence value with the same colors as in (**b**) and (**c**). **b** Residual rate map in which the mean temporal trajectory is subtracted. **c** Tuning curve averaged over time ($n = 33$ time bins). Points indicate mean (±s.e.m.) across time of the change in firing rate relative to temporal average as a function of accumulated evidence value $a$. **d** Rank 1 approximation of the residual rate map $E[\Delta r|a, t]$ from (**b**). The approximation (left) is equal to the outer product of a modulation over time $\hat{m}(t)$ (middle) and a tuning curve $\hat{r}(a)$ (right). **e** Average residual $z$ scored firing rate map (left). This plot is produced by averaging over the residual $z$ scored firing rate map of all pre-movement side-selective cells. This map is approximated by the outer product of a modulation curve (middle) and a tuning curve (right). Source data are provided as a Source Data file.

decision. Unlike the previous analysis (Fig. 3), here we isolate time points around model-predicted changes of mind, grouping data only by the inferred provisional decision. This approach allows us to confirm that the stable choice coding seen in Fig. 3 is not an artifact produced by averaging in a subset of trials with stronger coding and fewer changes of mind.

To look at responses to model-predicted changes of mind, we computed each cell's average deviation from its mean temporal trajectory aligned to time points when the behavioral model predicted a change in the animal's estimate of the environmental state (Fig. 4a). Following Hanks et al.[7], we introduced a 100 ms response lag between model-predictions and FOF responses. For this analysis, model state changes were selected at time points when a 100 ms running average of the posterior mean crossed the decision boundary $B$. To avoid introducing noise into this analysis, model state changes in the first and last 200 ms of the trial were excluded, as were state changes that immediately reversed to the previous state (see methods). For each cell, this method produced two state-change triggered response curves

describing responses to changes into states 1 ($STR_1$) and 2 ($STR_2$). STRs are also referred to as $STR_{pref}$ and $STR_{non-pref}$ according to cells' previously determined side-preference. STRs are shown for an example neuron (Fig. 4b). Discriminability before and after model state changes was measured using d' and tested for significance by permuting the state-change labels across trials (2-tailed permutation test, 250 permutations, $p < 0.05$).

To visualize the state change triggered response across the neural population, each cell's response is summarized by computing the difference between the $z$ scored STR for state changes into the preferred state and into the non-preferred state ($STRpref - STRnon - pref$). We present these data as a heat map for all pre-movement side-selective cells (Fig. 4c). The $z$ scored STRs were averaged across these cells to give the average state-change triggered response across the population (Fig. 4d). We apply the permutation procedure described above to each cell and compute the fraction of the included cells that significantly encode state at each timepoint relative to the state change (Fig. 4e). If cells are encoding the animal's provisional decision,

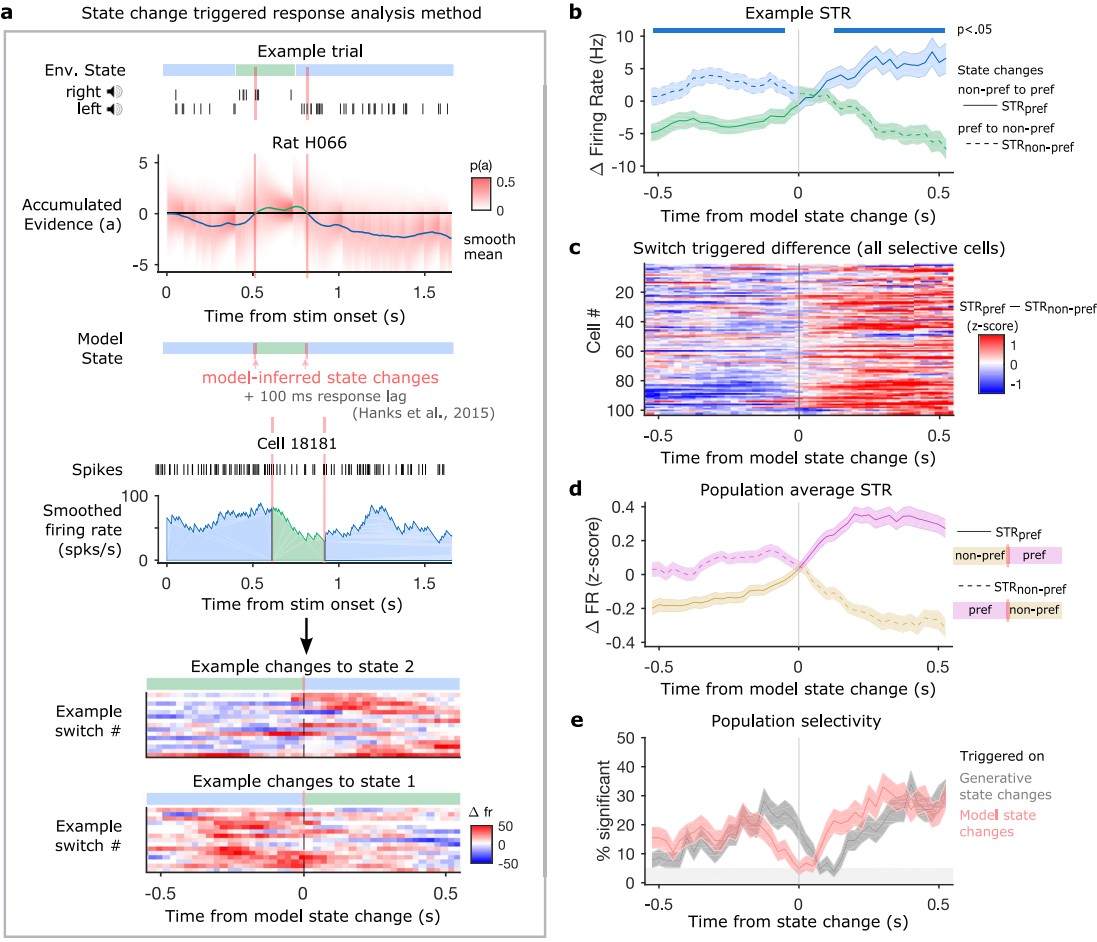

**Fig. 4 FOF neurons track changes in the provisional decision. a** Schematic explaining method used to compute state change triggered responses (STR). A given trial has a hidden environmental state (blue and green bar) used to generate click trains from each speaker. We compute the posterior distribution of accumulated evidence given the choice at each time point, $p(a)$. We find time points where the smoothed posterior mean crosses the decision boundary and label these model-inferred state changes. We then select the residual smoothed firing rates from the 550 ms before and after each state change and average together the residual responses for changes into state 1 and changes into state 2. **b** STR (mean ± s.e.m.) for the example cell used in panel A. Significance bars indicate time points when $d'$ for discriminating model state is different from chance (2-tailed permutation test, 250 permutations, $p < 0.05$). The trace showing changes into the cell's preferred state (state 2 for this cell) is labeled $STR_{pref}$ (solid line) and the trace for changes into the cell's non-preferred state is labeled $STR_{non-pref}$ (dashed line). **c** Heat map showing difference between responses for changes into the preferred and non-preferred state ($STRpref - STRnon-pref$) for each of the pre-movement side-selective cells. **d** Average $z$ scored STR (mean ± s.e.m.) across all pre-movement side-selective cells for state changes into cells' preferred states and non-preferred states. **e** Percentage of included cells (mean ± s.e.m.) with significant encoding across time relative to model predicted state changes (red trace) and generative state changes (gray trace). Source data are provided as a Source Data file.

we expect them to take intermediate firing rates during changes of mind and not show significant encoding of either state. If our behavioral model accurately predicts the timing of changes of mind, these intermediate firing rates should coincide with model state changes. As predicted, we find that the population reaches its minimum fraction of cells differentiating between states at the time of the model-predicted state change. We recomputed the timecourse of discriminability across cells triggered on changes in the veridical environmental state, rather than the model-predicted changes. When we do this, we find the time point at which the minimum fraction of cells significantly discriminates between states is delayed relative to generative state changes. This is consistent with the FOF tracking changes in the sign of accumulated evidence rather than simply responding to the instantaneous stimulus. At the level of individual cells and across the population, we see rapid responses to changes of mind, providing further evidence that neurons track the animal's provisional decision throughout the accumulation process.

**Stable evidence tuning before and after changes of mind**. To further characterize cell tuning to accumulated evidence during changes of mind, we recomputed the tuning maps aligning time to model-predicted state changes instead of the start of the trial. This analysis is restricted to the time points around model state changes as in Fig. 4, but also allows us to more closely examine the stability of tuning before and after these events. The computation and rank 1 decomposition of the tuning curves proceeded in the same manner as before except time in each trial was aligned to model state changes:

$$r(a, t - t_c) \approx E[r|t] + \hat{m}(t - t_c) * \hat{f}(a) \qquad (4)$$

where $t_c$ is the timing of model state changes. Consistent with the state-change triggered responses and previous tuning curve analysis, we see that tuning in example neurons and the population is well described by a single evidence tuning curve multiplied by a temporal modulation before and after state changes (Fig. 5a, b). The rank 1 approximation for the example cell presented in

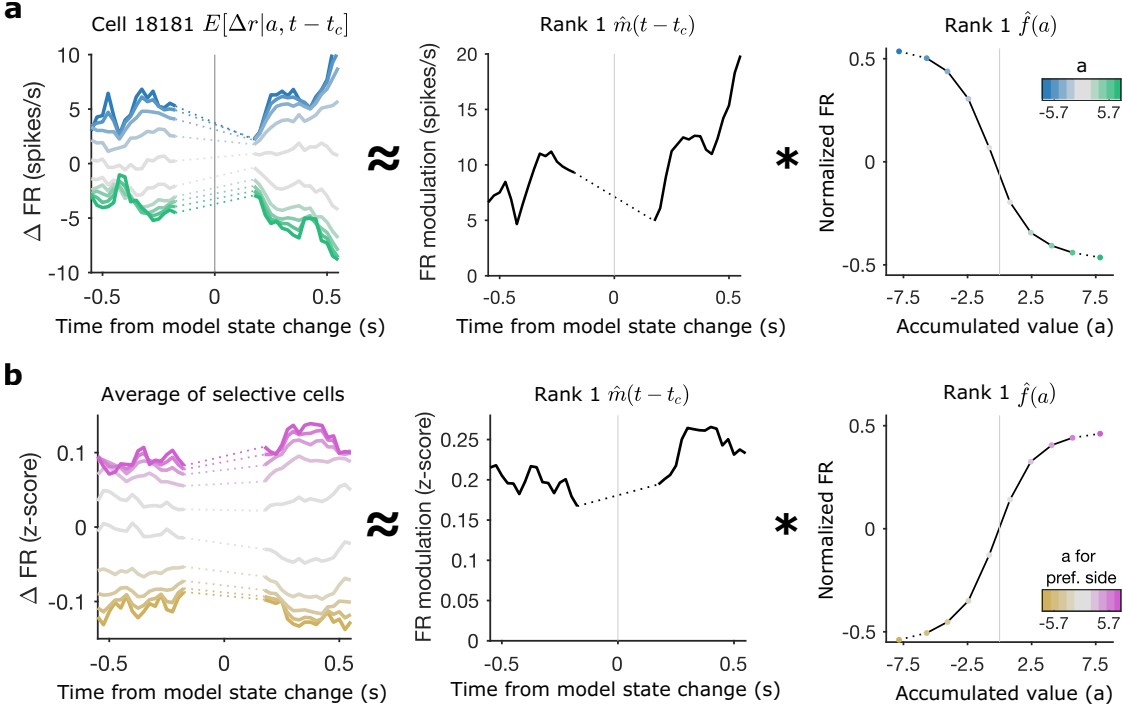

**Fig. 5 Stable tuning curve captures responses before and after model state changes. a** Example cell tuning map triggered on model-predicted state changes with rank 1 approximation derived temporal modulation and evidence tuning. Data is excluded from the 300 ms around the state change where the accumulated value distribution is too narrow to estimate tuning (dotted lines). **b** Average of all pre-movement side-selective cells' tuning maps computed with z scored firing rates and triggered on model-predicted state changes along with rank 1 approximation derived temporal modulation and evidence tuning for the average map. Source data are provided as a Source Data file.

Fig. 5a explains 98.9% of the variance in the tuning map and the average variance explained for all selective cells is 84.9% ± 9.8%. The population average across z scored tuning maps for all pre-movement side-selective cells is also well-described by the rank 1 approximation, which captures 89.2% of the variance (Fig. 5b). This demonstrates that neurons encode the accumulated evidence with a single tuning curve even at the times when the hidden state and provisional decision fluctuate.

## Discussion

We recorded neural activity from the frontal orienting fields (FOF) of rats performing a dynamic decision-making task designed to induce frequent changes of mind. In our study, rats integrated sequential pieces of information, discounting older evidence, to track changes in a volatile hidden state. FOF responses have been characterized previously during a similar task in a stationary environment where rats learn to equally weigh all evidence and changes of mind are rare[7]. This previous work revealed categorical encoding of population activity to the accumulated evidence, characterized by a single tuning curve throughout the trial. This suggested that FOF encoded the provisional decision during evidence accumulation. However, in a stationary environment, the provisional decision rarely differs from the final choice meaning that preparatory activity could begin without needing to be reversed. In a dynamic environment, where changes of mind are frequent, it might be advantageous to suppress choice coding until the final decision is reached. It was not clear whether FOF would play a similar role in representing evidence during decision-making in a constantly-changing environment and while the provisional decisions were still highly flexible.

We found that FOF responses to accumulation in a dynamic environment were similar to FOF responses during accumulation in a stationary environment. First, a subpopulation of about 18% of active neurons showed significant side-selectivity during the pre-movement stimulus period. This was a smaller fraction than previously reported, but was an expected result of a task with more frequent stimulus-induced changes of mind. Using a method developed by Hanks et al.[7], we measured the encoding of the decision variable in single neurons and across the population. We improved this method by using a rank 1 approximation to explain the evidence-encoding component of neural firing rates as the product of a temporal modulation and an evidence tuning curve. The rank 1 approximation supported the description of FOF neurons with a single evidence tuning curve that was modulated over the trial. Across the population, we found that the temporal modulation increased until the timing of the earliest possible "go" cue and then plateaus at a maximum modulation strength during the rest of the trial.

The dynamic nature of the task allowed measurements that are not possible in stationary tasks, where evidence is drawn from a single distribution during each decision, and changes of mind are rare. We used our behavioral model to estimate the rat's provisional decision throughout each trial. Fluctuations in this model state variable provided an estimate of the timing of changes of mind for analysis of neural activity. If the neurons use a single evidence tuning curve throughout accumulation, we expect the neural firing rates to encode the provisional decision before and after changes of mind. Computing state change triggered responses for each neuron, showed that FOF cells responded rapidly to model state changes, reflecting the new provisional decision in their firing rates. Critically, neurons encoded provisional decisions both before and after these events, which implies that provisional decisions are encoded even when they differ from the final choice. Neuronal responses were better aligned to state changes predicted by the behavioral model than to changes in the

true environmental state, suggesting that these responses were not simply reflecting a change in sensory experience. Combining this approach with the method for computing accumulated evidence tuning maps, we found, as described above, that the product of a single evidence tuning curve and temporal modulation was still sufficient to explain the evidence response across model state changes (rank 1 approximation). We observed that, after the moment when "go" cues could arrive, the temporal modulation of evidence tuning was, on average, stable. Together, our results demonstrate that FOF neurons encode the animal's provisional decision and respond rapidly, updating this representation following changes of mind.

One important limitation of our study is that our evidence accumulation model only uses one fixed set of parameters to describe each rat's behavior in a trial in terms of the stimulus on that trial. While the model is highly flexible and captures average behavior, it does not allow parameters to change over trials, nor does it capture trial-to-trial history effects. Future work should develop more flexible behavioral models to capture slow drifts and sudden state changes in the parameters that describe the animals' strategies. This work will allow deeper investigation into neural coding.

Changes of mind are not unique to dynamic environments and can also occur during evidence accumulation in stationary environments. These events can occur during stimulus presentation due to noise in the decision making process and can be predicted from neural activity[25]. Changes of mind may also occur after the subject begins to execute their choice due to post-processing delays[21] or constraints placed on action[26]. Our work differs from these studies, in that we use an environment designed to induce changes of mind and ask how neurons respond to these model-predicted events. To our knowledge, only one other study[27] has examined neural responses to behaviorally predicted changes of mind during evidence accumulation in a dynamic environment, and ours is the only such study in an animal model.

Previous inactivation studies suggest that while FOF is critical for performing actions and reporting decisions, it is not necessary for the integration of evidence[7,28]. This is consistent with the FOF representing the evidence after categorization into a provisional choice[14]. Work in mouse anterior lateral motor (ALM), a comparable cortical region, shows that categorical signals in this region recover quickly following photoinhibition, suggesting categorical input from other brain regions[29]. In a recent study, Finkelstein et al.[30] found that ALM choice signals were robust to distractors delivered during a delay period after the typical evidence presentation period, suggesting local circuitry maintained the choice signal. Our study considered a similar brain structure operating in a regime where, rather than ignoring distractors, it needed to flexibly update provisional decisions in response to new information. These studies, along with recent modeling work[14,31], suggest a common role for the FOF and the ALM in maintaining choice signals that are either robust to or responsive to new information according to task demands.

The dynamic decision-making task offers a complementary approach to typical studies of evidence accumulation in static environments. Here, we showed that in constantly-changing environments FOF neurons encode provisional choices and respond rapidly to changes of mind predicted from our behavioral model. Our quantitative methods and behavioral paradigm will be useful tools for investigation of the brain circuitry supporting evidence accumulation and the decision-making process more generally.

## Methods

**Subjects**. Animal use procedures were approved by the Princeton University Institutional Animal Care and Use Committee and carried out in accordance with NIH standards. All subjects were adult male Long Evans rats (Vendor: Taconic,

Hilltop and Harlan, USA). Rats were pair-housed prior to implantation with recording electrodes and single-housed subsequently. Rats were placed on a water restriction schedule to motivate them to perform the task for water rewards.

**Behavioral training**. We trained rats on the dynamic clicks task[18] (Fig. 1). Rats went through several stages of an automated training protocol. In the final stage of training, each trial began with the illumination of a center nose port by an LED light inside the port. This LED indicated that the rat could initiate a trial by placed its nose into the center port. Rats were required to keep their nose in the center port (nose fixation) until the light turned off as a "go" signal. During center fixation, auditory cues were played indicating the current hidden state. The duration of the stimulus period was drawn from a uniform distribution between 500 and 2000 ms. After the "go" signal, rats were rewarded for entering the side port corresponding to the final value of the hidden state. The hidden state did not change after the "go" cue. Correct choices were rewarded with 18 microliters of water. Incorrect choices were signaled by a white noise stimulus (spectral noise of 1 kHz for a 0.7 s duration). The rats were put on a controlled water schedule where they receive at least 3% of their weight every day. Rats trained each day in training session of around 120 min. Training sessions were included for analysis if the overall accuracy rate exceeded 70%, the center-fixation violation rate was below 25%, and the rat performed more than 50 trials. In order to prevent the rats from developing biases towards particular side ports an anti-biasing algorithm detected biases and probabilistically generated trials with the correct answer on the non-favored side.

**Psychometric and chronometric curves**. Task performance was assessed using psychometric curves, chronometric curves and psychophysical reverse correlations. For all task performance plots, rat data was overlaid on predictions from the accumulation model described below. These predictions were made by using the probability of a right or correct choice on each trial given by the acummulation model in place of the actual choice observed.

Psychometric plots show the probability that the rat chose to go right as a function of the ideal observer log-odds supporting a "go right" choice. Final state chronometric plots show the probability of a correct choice as a function of the final state duration, the elapsed time between the final hidden state change (or the beginning of the stimulus, if there are no state changes) and the end of the stimulus. Data is plotted separately for trials with 0, 1, or more than 1 state changes.

**Psychophysical reverse correlation**. The computation of the reverse correlation curves was similar to methods previously reported[7,8,28]. An additional step was included, as in Piet et al.[18], to deal with the changing hidden state. First, the right and left click trains were each smoothed using a causal Gaussian filter $k$ with a standard deviation of 5 msec. The smoothed left clicks were then subtracted from the smoothed right clicks, creating one smooth click difference rate $d$ for each trial:

$$d(t) = (\delta_R * k)(t) - (\delta_L * k)(t). \tag{5}$$

Here, the click train $\delta_R$ is a sum of delta functions with peaks at the time of each right click and the value 0 everywhere else. Then, the expected click difference rate given the current state of the environment, $E[d(t)|S(t)]$, was subtracted from $d$ at each timepoint on each trial. Here, $S(t)$ is the current environmental state. This gives us the deviation from the expected click difference rate for each trial. This is called the excess click difference rate or just the excess click rate.

$$e(t) = d(t) - E[d(t)|S(t)] \tag{6}$$

Finally, we compute the choice-triggered average of the excess click rate by averaging over trials conditioned on the rat's choice $y \in \{-1, 1\}$.

$$\text{excess-rate}(t|y) = E[e(t)|y] \tag{7}$$

The excess rate curves were then normalized to integrate to one. This was done to remove distorting effects of a lapse rate, as well to make the curves more interpretable by putting the units into effective weight of each click on choice.

**Accumulation model**. The accumulation model characterizes the decision-making process as the evolution over time $t$ of an accumulation value $a$ in response to left and right click trains, $\delta_L$ and $\delta_R$, with dynamics governed by a parameter set $\theta$. Each rat's behavioral data is used to find the parameter set that maximizes the probability under the model of the rat's choices $y$. Evaluating this model with the best fit parameters produces a probability distribution over values of $a$ at every timepoint in the trial. We refer to this as the forward model distribution $f(a) = P(a|t, \delta_R, \delta_L, \theta, a_0 \sim \mathcal{N}(0, \sigma_I^2))$. The forward model was described previously in Piet et al.[18] and will be reviewed in detail below. To characterize neural encoding of the accumulation value, we further constrained the accumulation value distribution on trials where we had simultaneous neural recordings by incorporating the rat's choice, $y$, to find the posterior distribution $p(a) = P(a|t, \delta_R, \delta_L, \theta, a_0 \sim \mathcal{N}(0, \sigma_I^2), y)$. To do this, we computed a distribution that we refer to as the backward model distribution, which we describe in the next section.

The accumulation model is a stochastic differential equation that describes the evolution of an accumulation value $a$, and a sensory adaptation value $C$:

$$da = \left(\delta_{R,t} \cdot \eta_R \cdot C - \delta_{L,t} \cdot \eta_L \cdot C\right)dt - \lambda a dt + \sigma_a dW, \quad (8)$$

$$\frac{dC}{dt} = \frac{1-C}{\tau_\phi} + (\phi - 1)C\left(\delta_{R,t} + \delta_{L,t}\right). \quad (9)$$

Each sensory click is scaled by the sensory adaptation value C and multiplicative Gaussian noise $\eta$ drawn from $\mathcal{N}(1, \sigma_s^2)$. The model parameters $\theta$ can be described in words as an initial noise variance $\sigma_i^2$, a per-click noise variance $\sigma_s^2$, a memory noise variance $\sigma_a^2$, a discounting rate $\lambda$, the strength and time constant of adaptation $\phi$ and $\tau_\phi$, a decision boundary $B$, which captures the animal's bias, and a lapse rate $l$. If the adaptation strength parameter $\phi < 1$, then consecutive clicks are depressed. If $\phi > 1$, then consecutive clicks are facilitated.

The forward model is the solution to Eqn. (8), assuming the initial accumulation value $a_0$ is Gaussian distributed with zero mean and variance $\sigma_i^2$. At each moment in the trial, the forward model $f(a) = P(a|t, \delta_R, \delta_L, \theta, a_0 \sim \mathcal{N}(0, \sigma_i^2))$ predicts a Gaussian distribution of accumulation values with mean $\mu(t)$ and variance $\sigma^2(t)$ given by:

$$\mu(t) = \mu_0 e^{\lambda t} + \int_0^t \left(\delta_{R,s} \cdot C(s) - \delta_{L,s} \cdot C(s)\right)ds$$
$$= \sum_i^{\#R_t} e^{\lambda(t-R(i))}C(R(i)) - \sum_i^{\#L_t} e^{\lambda(t-L(i))}C(L(i)) \quad (10)$$

$$\sigma^2(t) = \sigma_i^2 e^{2\lambda t} + \frac{\sigma_a^2}{2\lambda}\left(e^{2\lambda t} - 1\right) + \int_0^t \sigma_s^2\left(\delta_{R,s} \cdot C(s) - \delta_{L,s} \cdot C(s)\right)e^{2\lambda t}ds$$
$$= \sigma_i^2 e^{2\lambda t} + \frac{\sigma_a^2}{2\lambda}\left(e^{2\lambda t} - 1\right) + \sum_i^{\#R_t} \sigma_s^2 C(R(i))e^{2\lambda(t-R(i))} + \sum_i^{\#L_t} \sigma_s^2 C(L(i))e^{2\lambda(t-L(i))} \quad (11)$$

Where $\delta_{R,t}$ indicates whether there was a right click at time $t$ and $C(t)$ tells us the effective adaptation for a click at time $t$. For the discrete case, $\#R_t$ is the number of right clicks on this trial up to time $t$ and $R(i)$ is the time of the $i^{th}$ right click.

To determine the probability of a right versus left choice, we first integrate the accumulation value distribution in the last timepoint $t_N$ of the trial from the decision boundary parameter $B$ to $\infty$

$$P(a>B|t = t_N, \delta_R, \delta_L, \theta, a_0 \sim \mathcal{N}(0, \sigma_i^2)) = \frac{1}{2}\left(1 + \text{erf}\left(\frac{-(B - \mu(t_N))}{\sigma(t_N)\sqrt{2}}\right)\right). \quad (12)$$

On each trial, the rat makes a random choice with probability determined by lapse rate $l$. Then, the probability of a "go right" choice is given by

$$P(y = 1|\theta) = (1-l)P(a > B|t = t_N, \delta_R, \delta_L, \theta, a_0 \sim \mathcal{N}(0, \sigma_i^2)) + l/2 \quad (13)$$

$$P(y = -1|\theta) = (1-l)\left(1 - P(a > B|t = t_N, \delta_R, \delta_L, \theta, a_0 \sim \mathcal{N}(0, \sigma_i^2))\right) + l/2 \quad (14)$$

Where

$$y = \begin{cases} 1, & \text{if rat chooses right} \\ -1, & \text{if rat chooses left} \end{cases} \quad (15)$$

Parameters $\theta$ were fit to each rat individually by maximizing the likelihood function:

$$L = \prod_i^{\#\text{trials}} P(y^i|\theta). \quad (16)$$

A half-Gaussian prior was included on the initial noise $\sigma_i^2$ and accumulation noise parameters $\sigma_a^2$. The priors were set to match the respective best fit values from Brunton et al.[8]. The numerical optimization was performed in MATLAB, using the function `fmincon`. To estimate the uncertainty on the parameter estimates, we used the inverse hessian matrix as a parameter covariance matrix[32]. To compute the hessian of the model, we performed automatic differentiation in *julia* to exactly compute the local curvature[33]. See the Supplementary Information for parameter estimates and uncertainty values. Brunton et al.[8] extensively analyzed how well a similar model with an additional bound parameter recovers generative parameters, finding the model contains one maximum likelihood point in parameter space (See Section 2.3.3-6 of the Supplement to Brunton et al.[8]). We compared parameter fits in this task to those reported in Brunton et al.[8], which developed the stationary version of this task.

**Posterior model**. The forward model described above gives us a probability distribution over accumulation values at each time point in each trial. It also gives an estimated probability of the rat choosing to go right or left on that trial. Observing the rat's choice $y$ at the end of each trial allows us to constrain the distribution of possible trajectories that the accumulation value could have taken. The resulting posterior distribution (referred to as the backward pass distribution in Brunton et al.[8]) is useful for analyzing the neural encoding of accumulated evidence.

We develop a novel method of computing the posterior distribution by taking the product of the forward distribution and a backward distribution. Again, we note that while Brunton et al.[8] refers to the posterior distribution as the backward pass distribution, we use the term backward distribution to refer to a distinct distribution which constrains the final state of the accumulation value distribution in accordance with the animal's choice, but does not constrain the initial state. As described above, the forward distribution assumes that the initial accumulation value $a_0$ is normally distributed with mean 0 and variance $\sigma_i^2$. The backward distribution makes no assumption about the initial distribution, but assumes that the final accumulation value $a_N$ is uniformly distributed on the side of the decision boundary $B$ that corresponds to the rat's choice. Importantly, the forward and backward distributions are conditionally independent, conditioned on the final value of the accumulated evidence. Given that these distributions are independent, their product gives the posterior distribution $p(a)$ that combines the constraints on the initial and final distributions of accumulation values:

$$p(a) = P(a|t, \delta_R, \delta_L, \theta, a_0 \sim \mathcal{N}(0, \sigma_i^2), y) \quad (17)$$

$$\propto f(a)b(a). \quad (18)$$

Where $f(a)$ is the forward model described above, which assumes a Gaussian initial distribution of accumulation values depending on $\sigma_i^2$:

$$f(a) = P(a|t, \delta_R, \delta_L, \theta, a_0 \sim \mathcal{N}(0, \sigma_i^2)) \quad (19)$$

and $b(a)$ is the backward distribution, which assumes the final accumulation value is on the side of the decision boundary $B$ corresponding to the animal's choice $y$:

$$b(a) = P(a|t, \delta_R, \delta_L, \theta, y) \quad (20)$$

$$= \begin{cases} P(a|t, \delta_R, \delta_L, \theta, a_N \geq B), & \text{if } y = 1 \\ P(a|t, \delta_R, \delta_L, \theta, a_N \leq B), & \text{if } y = -1. \end{cases} \quad (21)$$

We approximated the backward distribution as a mixture distribution over a grid of final accumulation values with spacing $\Delta a$. A unit of probability mass is initialized at each point in the grid and the solution is given by:

$$b(a) = \sum_{j=0}^{\pm\infty} w_j P(a|t, \delta_R, \delta_L, \theta, a_N \sim \mathcal{N}(B + j\Delta a, 0)) \quad (22)$$

The mixture weights $w_j$ are all equal if the bin spacing is uniform. Each unit of probability mass evolves using the same solution as the forward model, but with time reversed. This solution is exact as $\Delta a \to 0$.

For tuning curve analyses we use the full posterior distribution, for the state change triggered response analyses we use the mean of the posterior. See the Supplementary Information for a detailed discussion on the derivation and evaluation of the backward and posterior model.

**Microwire array recordings**. Microwire array implant surgery: Four rats were implanted with microwire arrays in their left or right FOF ($n = 2$ in lFOF, $n = 2$ in rFOF) The target region was accessed by craniotomy, using standard stereotaxic techniques (centred 2 mm anterior to the bregma and 1.3 mm lateral to the mid-line). Dura mater was removed over the entire craniotomy with a small syringe needle. The remaining pia mater, even if not usually considered to be resistant to penetration, nevertheless presents a barrier to the entry of the microelectrode arrays because of the high-density arrangement of electrodes in the multi-channel electrode arrays. This dimpling phenomenon, when the electrodes are pushing the brain cortex down without penetrating, is more pronounced for arrays with larger numbers of electrodes. In addition to potentially injuring the brain tissue, dimpling is a source of error in the determination of depth measurements. Ideally, if dimpling could be eliminated, the electrodes would move in relation to the pial surface, allowing for effective and accurate electrode placement. To overcome the dimpling problem, we implemented the following procedure. After the craniotomy was made, and the dura was carefully removed over the entire craniotomy, a petroleum-based ointment (such as bacitracin ointment or sterile petroleum jelly (Puralube Vet Ointment)) was applied to the exact site of electrode implantation. The cyanoacrylate adhesive (Vetbond Tissue Adhesive) was then applied to the zone of the pia surrounding the penetration area. This procedure fastens the pia mater to the overlying bone and the resulting surface tension prevents the brain from compressing under the advancing electrodes. Once the polymerization of cyanoacrylate adhesive was complete, over a period of few minutes, the petroleum ointment at the target site was removed, and the 32-electrode microwire array (Tucker-Davis Technologies) was inserted by slowly advancing a Narishige hydraulic micromanipulator. After inserting the array(s), the remaining exposed cortex was covered with biocompatible silicone (kwik-sil), and the microwire array was secured to the skull with C&B Metabond and dental acrylic.

During a ten-day recovery period, rats had unlimited access to water and food. Recording sessions in the apparatus began thereafter, using Neuralynx acquisition systems. Once rats had recovered from surgery, recording sessions were performed in a behavioral chamber outfitted with a 32 channel recording system (Neuralynx). Spiking data was acquired using a bandpass filter between 600 and 6000 Hz and a spike detection threshold of 30 microV.

For array recordings, clusters were manually cut (Spikesort 3D, Neuralynx), and both single- and multi-units were considered.

**Tetrode recordings**. Tetrode drives were 3d printed from custom designs (design files available upon request) on a Form2 3d printer in tough resin. Each drive consists of a drive body, a cone and cap to protect the drive body, and four bundles of 8 tetrodes in glass tubes. Each bundle was glued together and to a cannula. Each cannula was attached to a screw using dental cement, and cured with UV light. Each wire from each tetrode was fed through a unique channel in a 128 channel Electrode Interface Board (SpikeGadgets) and pinned with a gold pin. After loading all tetrodes, trimming, and building of the drive, the day-of or night before the surgery, we electroplated the drive in gold using a nanoZ impedence tester (White Matter LLC) and measured impedences.

Tetrode drive implant surgery proceded as described for microwire arrays, except we did not need to vetbond the brain surface because each tetrode bundle produced very little dimpling. A silver wire and skull screw were used to ground the drive. Drives were secured with metabond and acrylic until secure. Tetrodes were advanced 0.1 mm into the brain.

During a seven-day recovery period, animals had unlimited access to water and food. Animals were then returned to training and water restriction. To acclimate animals to the weight of the wireless apparatus, every three days, we replaced the cap on the implant with a new cap 3 g heavier than the previous cap. If animals' behavioral performance or weight dropped, or if we noticed any excess tilting of the head from the weight, we returned the animal to the previous weight and waited an additional 2 days before moving to the next weight. This process was repeated until the animals were behaving well with caps weighing 27 g.

Once animals were acclimated to the weight, recordings could begin. Tetrodes were advanced 0.25 mm at a time, at least 20 h before recording. For each recording session, the animal's cap was replaced with a 500 mAh lithium battery, 128 Gb Sandisk extreme plus SD card, a 160-pin Amphenol Lynx connector, and datalogger (SpikeGadgets). At the end of each session, the datalogger, SD card, and battery were removed and the 27 g cap replaced.

The tetrode recordings were automatically clustered using Kilosort2[34]. Automatically determined clusters were manually curated using the Phy GUI (https://github.com/kwikteam/phy).

**Electrophysiological analysis**. We computed the firing rates for all neurons aligned to the time of stimulus onset (when the rat first broke the center port IR beam triggering playback of the stimulus) and to movement (when the rat first stopped breaking the center port IR beam to make its choice). Firing rates were computed by binning spikes into 25 ms bins and smoothing them with a casual Gaussian filter with a standard deviation of 100 ms. Stimulus onset aligned firing rates were masked on each trial after the movement and movement aligned firing rates were masked prior to stimulus onset. Firing rates for example cells were averaged over trials conditioned on choice and outcome.

Cells were considered active if their average stimulus onset aligned firing rate was greater than 1 Hz during the time from 1 s prior to the stimulus onset to the time of movement onset. Cells were considered pre-movement side-selective if the spike counts during the period between stimulus onset and movement were different on trials that resulted in a left versus a right choice (2-tailed $t$ test, $p < 0.05$). The side with the higher firing rate is referred to as the cell's preferred side.

A population-average PSTH was computed by averaging over all trials from all pre-movement side-selective cells conditioned on final state duration and whether the trial ended in a choice to the cell's preferred side.

We analyzed the timecourse of choice-selectivity by computing the area under the receiver operating characteristic curve (AUC) at each 25 ms time bin in for the smoothed firing rates in left choice versus right choice trials. In this application, the receiver operating characteristic curve (ROC) treats the spike rate in a time bin as a classifier of right versus left choice, computing the true positive rate and false positive rate as a function of the spike threshold. The area under this curve is equivalent to the probability that in a randomly chosen pair of right left choice trials the firing rate in that bin will be higher on the right choice trial than on the left choice trial. AUC values significantly greater than 0.5 indicate a preference for right choice trials and AUC values significantly less than 0.5 indicate a preference for left choice trials[35]. To compute significance, we performed a permutation test where the left/right choice labels were permuted relative to the firing rates across trials. For visualization purposes, we sorted cells by latency to reach 8 significant 25 ms time bins in a row (2-tailed permutation test, 250 permutations, $p < 0.05$).

**Evidence tuning curves**. We compute evidence tuning curves using a method based on the one used in Hanks et al.[7]. First, the posterior accumulation distribution $p(a)$ for each trial is computed, providing a distribution over the evolution of the accumulation values for each trial that is consistent with the rat's choice on that trial. The joint distribution of $p(a)$, the firing rate $r$, and time $t$, which we will call $P(r, a, t)$ is computed by binning time, accumulation values, and firing rates. For each trial and each timepoint, the probability mass in each accumulation value bin in $p(a)$ is added to the bin in $P(r, a, t)$ associated with that timepoint and that firing rate. Because the shortest trials are 500 ms, not all trials contribute to each time point, so each time bin is normalized according to the number of trials that contribute to it. When estimating

the joint distribution we discretized our data along three dimensions: time, firing rate, and accumulation value. Time was binned into 25 ms bins. Each neuron's firing rate was divided into 100 bins spanning the minimum to the maximum firing rate of the neuron. The firing rate bin was chosen by taking the average firing rate within the time bin. Accumulation value bin size was divided into 10 bins with width set to 1.625 except the last bin which was larger to capture the tails of the distribution. The posterior distribution was evaluated with 1 ms time bins and accumulation value bin size of 0.1 and then downsampled to populate the joint distribution.

To estimate each cell's firing rate map, we compute the conditional expectation of the firing rate in each accumulation value and time bin $E[r|a, t] = \sum_r rP(r|a, t)$. We then computed the expected difference, at each time point, from the cell's time-averaged firing rate as a function of the accumulation value $E[\Delta r|a, t] = E[r|a, t] - E[r|t]$, where $E[r|t]$ is the average of $E[r|a, t]$ over all values of a. Average accumulation value tuning over the full trial is computed by averaging $E[\Delta r|a, t]$ over time to get $E[\Delta r|a]$. The same procedure is used to compute a map of $z$ scored firing rates. And these maps are averaged across pre-movement side-selective cells to produce a population average.

Rank 1 approximations of the $E[\Delta r|a, t]$ are computed using the singular value decomposition. The approximation is equal to the outer product of the first left singular vector $u_1$ and the first right singular vector $v_1$, scaled by the first singular value $s_1$. These terms are rearranged to give the outer product of a firing rate modulation, $\hat{m}(t) = u_1 s_1 \text{ range}(v_1)$ and a tuning curve $\hat{f}(a) = v_1 / \text{range}(v_1)$. Scaling by $\text{range}(v_1)$ gives $\hat{f}(a)$ unit scale and $\hat{m}(t)$ units of spikes/s. Our complete tuning curve approximation becomes:

$$r(a, t) \approx E[r|t] + \hat{m}(t) * \hat{f}(a). \tag{23}$$

The variance explained by this approximation is given by the ratio between the first singular value and the sum of all singular values:

$$\frac{s_1}{\sum_i s_i}. \tag{24}$$

**State change triggered response**. On each trial, we computed a residual firing rate by subtracting the average firing rate at each time point during the trial. We then aligned these residual firing rates to either the model-predicted state changes or the generative environmental state changes. We masked firing rates before the preceding state change and after the following state change when applicable. We computed the mean of these residual firing rates for visualization. To test for significant discrimination of state, we compared $d'$ in the real data to a permutation distribution created by permuting state labels across state changes (2-tailed permutation test, 250 permutations, $p < 0.05$).

We defined model-predicted state changes as time points where the running average of the mean of the posterior accumulator value crossed the rat's best fit decision boundary $B$. The running average was computed over 100 bins of 1 ms. To avoid introducing noisy state changes, we excluded state changes from the first and last 200 ms of the trial. We also excluded state changes that did not meet two change strength criteria designed to identify state changes that were immediately reversed. The first, was based on the average value of the posterior mean in the 100 ms before the change compared to the 100 ms after the change. State changes were excluded if these strength values were inconsistent with the direction of the identified state change. The second state change strength was based on the slope of the running average of the posterior mean at the time of the change. If the sign of the slope was inconsistent with the sign of the state following the state change, this means that the accumulation value immediately returned back to the previous state. We excluded these state changes. Our results were robust to variations in the state change inclusion criteria.

**State change triggered tuning**. State change triggered tuning maps $E[\Delta r|a, t - t_c)]$ were computed using the tuning curve methods described above, but using time relative to state changes instead of stimulus onset. Firing rates were masked before and after the preceding and following state changes as described above. Data was also masked in the 300 ms around the state change where the accumulated value distribution is too narrow to estimate tuning. Rank 1 approximations and population tuning maps were computed as above.

**Reporting summary**. Further information on research design is available in the Nature Research Reporting Summary linked to this article.

## Data availability
The data used in this paper are available at the following url: https://figshare.com/articles/dataset/Manuscript_Data/16695592. In addition, Source Data are provided with this paper, which can be used to reproduce figures without rerunning analyses. Source data are provided with this paper.

## Code availability
Analysis code used in this study is in the repository available at https://github.com/Brody-Lab/dynamic_ephys[36].

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

## Acknowledgements

We thank members of the Brody lab and Zachary Kilpatrick for useful conversations and feedback. T.B. acknowledges support by NIH grant T32 MH 65214-16. A.E.H. acknowledges support by NIH grant 1R21MH121889-01. E.J.D. is supported by an HHMI Hanna H. Gray Fellowship and an HHMI-Helen Hay Whitney Postdoctoral Fellowship. This work was supported by a grant from the Simons Foundation (Grant # 542953) awarded to C.B., as well as NIH grant R01MH108358 awarded to C.B.

## Author contributions

A.P., and A.E.H. designed the study. A.E.H. managed rat training and care. A.E.H., and E.J.D. recorded the neural data. A.P., and T.B. analyzed the neural data. A.P., A.E.H., and T.B. wrote the manuscript. A.E.H., and C.B. oversaw all aspects of the project.

## Competing interests

The authors declare no competing interests.
