## [Peer Review File · Nature Communications]

Stable choice coding in rat frontal orienting fields across model-predicted changes of mindREVIEWER COMMENTS

Reviewer #1 (Remarks to the Author):

In this work, Boyd-Meredith et al. examine neural activity in the frontal orienting field (FOF), which has previously been shown to track accumulated evidence for a two-alternative auditory decision task. Here, they consider a non-stationary version of the same task that has been previously modeled and ask whether single neurons continue to reflect these decision variables in cases where evidence may not necessarily rise uniformly during the task. They argue that this is indeed the case, that deviations from mean firing across trials may be modeled as a separable function of time and accumulated evidence, and that the timing of population responses matches the timing of statistical inference as captured by the model.

There is a great deal to like about this paper. The presentation is clear; I found the analyses well described and results compellingly laid out. The work continues a line of research from these investigators in an interesting and relevant direction and demonstrate the power of modeling tightly linked to neural physiology. I have some concerns, though, most related to presentation and nuancing results, but at least one serious.

Big picture things:

1. Despite some familiarity with the authors' previous work, I found the introduction a little narrowly focused and perhaps assuming a higher level of familiarity with recent FOF literature than a general reader might possess.

2. At several points, I would have appreciated more help with understanding possible alternatives to the authors' findings. For instance, given that the population of selected cells is chosen based on side responsiveness (essentially the difference between the means of the blue and green traces in Fig2A), it would have been good to know what possibilities this precludes. For instance, traces that oscillate or are u-shaped in time would presumably be missed by the selection criterion, which means that responses across trials in the surviving neurons should be either constant or roughly monotonic, correct? Likewise, for the rank-1 decomposition in (1), a separable rate function is clearly not the most general possibility, so what is the significance of the fact that this works? What would need to have been different for it to fail? Same thing for the analyses in Figure 4E and Figure 5. I *think* I understand the arguments here, but more discussion of how these relate back to the underlying scientific questions (especially with regard to possible alternatives) would be helpful.

Likewise II. 236-238: Given that firing rates are smoothed, and cells have been selected to have differencing firing rates by side, how can this fail? The assumption is that, if provisional decisions are reflected, then firing rates should transition between high and low as the evidence favors preferred versus non-preferred sides, correct? This may be what the authors are trying to say, but the point about an intermediate value needing to obtain between a high and low one seems to me trivial and perhaps not the clearest statement of the point.

3. Notation: quibbles about notation are petty at best, but (1) the authors have generally been careful and precise, so I hope they will take this in that spirit; and (2) in a couple of cases, it did cause me real confusion.

II. 543-547: Marginalization is an odd term here. Is this really, as written, $E[r|a,t] P(a, t)$? Why not use the conditional expectation? Similarly, the formula on I.545 is not $E[r|t]$ but $E[r|t]P(t)$. I would think that what the authors want is the conditional expectation (the average firing rate in each bin), but there seems to be some discrepancy between that and what they've written here. I'm unclear which they actually did.

On line 543, do they really mean $\sum_i r_i P(r_i|a, t)$? Similarly, in (2) and (18), is $E[r(t)]$ supposed to be $E[r|t]$?

Similarly in Figure 3, across the top, I would have expected to see $E[r|a, t]$, $E[r|a, t] - E[r|t]$, and $E[r|a] - E[r]$. It's not the Delta notation that I find problematic so much as the fact that it's unclear in the formulae which variables the expectation is being computed with respect to (plus my confusion about $E[r(a,t)]$ stated above).

The authors mix notation throughout the paper, sometimes using E for expectation, sometimes $\langle \rangle$ and $\langle \rangle$ (in (4) and (5)), and it would help readers, I think, to be consistent.

4. I have two confusions here that may either be misunderstandings or serious concerns:

4.1: On II. 437-440, it is stated that "the forward and backward distributions are conditionally independent, conditioned on the final value of the accumulated evidence." I *think* what the authors mean here is stated mathematically in (15). However, unless I am mistaken, the past and future distributions of the variable a are not independent unless one is also given the value of C . That is, unless one knows the value of $C(t)$, the process is no longer Markov, since one is missing the filtered information about the past contained in C .

4.2: In (14), the authors condition on y to produce the backward posterior $b(a)$. However, instead of conditioning on the set $[a_N \in [B, \infty)]$ for $y=1$, they additionally assume a uniform marginal distribution on the set. I have no idea why this should be true. More correctly, it can't be --- not just because the uniform distribution here is not normalizable, but because the marginal distribution of their stochastic process with Poisson increments should be Gaussian with a variance set by (24) and (25). Indeed, one should calculate the backward distribution by starting with the correct marginal for a_N , which is the approach originally used in Brunton, Botvinick, and Brody (for a discretized a). It is not at all to me clear how (28) and (29) approximate this unless the w_i are *not* all the same but are instead the marginal probabilities of a_N being in the given mixture component. I am not sure whether this affects the results of the paper, but this approach seems to me incorrect.

Minor points:

1. I. 47: This may not mean much to those less familiar with the FOF literature.
2. II. 62-63: This presumes that the animals are performing the hypothesized strategy. It may be cleaner to simply talk about static vs. dynamic environments and separate what is experimenter-controlled and what is results.
3. I realize the answer is in previous papers, but the equation governing C is never given.

Reviewer #2 (Remarks to the Author):

This manuscript can be viewed as a followup to the study of Hanks, et al 2015 (reference 6). Using similar methods the authors show that the activity of neurons in FOF represent provisional decisions in their firing rates even when changes in the environment lead to a change of mind.

The manuscript is interesting, and well written. The analysis is largely based on that in previous work by this group, but offers several extensions, and confirms previous findings in a more ethologically relevant setting. I believe the results and the approach are solid. I have some comments on the presentation, and clarifying questions the answers to which may help a general reader as well.

- Figure 3 is the highlight of the paper, but I found it confusing when returning to it after reading further. First, this demonstrates that the responses in FOF are categorical. However, the first and last column show graded responses. This argument is clear in light of the Hanks, et al 2015 paper, where FOF responses are compared to those of different areas. However, it should be clarified here.

What confused me further is whether this was data from the entire trial and thus averaged over changes in environment? This should be explained clearly - and contrasted to how the data was analyzed in Figs. 4 and 5.

- The description of the behavior model starting on line 99 could be a clearer. For instance, what does “this distribution” on line 103 refer to? The authors use a discrete time model, but the task occurs in continuous time. A lot of this is clear if a reader is familiar with the authors’ previous work. However, a novice reader will be confused.

- Following on the previous remark, the authors need to carefully check notation. First, some variables are overloaded: r stands for rate in the main text, but also for the filtered clicktrain in the methods. B stands for bias and decision boundary and bias. Most importantly, some quantities sometimes come with subscripts, and sometimes do not - for example a_0 and a . The subscripts are time bin indexes for some variables, but denote neuron number elsewhere. Some indices are not defined - eg I assume that a_N denotes the value of a in the last bin, N . However, I did not see N defined. A reader should not have to figure this out, nor have to go to previous papers to do so.

- A related point: Is B really a decision bound? The methods only mention the bias. I think they are equivalent in this context, but it may be best to keep the nomenclature consistent.

- A more general question - The authors assume that the rats use a normative model. This is consistent with Piet, et al. However, can simpler models be eliminated? For instance, the model where the rat goes with the side where it heard the last click or pair of clicks?

- Fig. 2A - It would be informative to see the variability in the responses, represented as an envelope, or by a sample of traces shaded more lightly overlaid over the average. A supplementary figures is fine.

- “Recordings from 69 sessions yielded 738 units.” - for all animals?

- Please explain the AUC computation better, either in the methods or supplement. The other techniques are explained well (modulo issues with notation described above), so it will be good to have a self-contained manuscript.

- The paragraph starting on line 176 is central, but it is a bit tough to follow. This is partly due to the notational issues mentioned above. however, I suggest carefully rewriting this paragraph to improve clarity.

- line 220 - decision boundary us defined as $a = 0$. What about bias?

- eq (3) - r_i , notation overloaded

- eq 13 and 14 - conditioning on $a_0 = \dots$ and $a_N = \dots$ I have not seen this notation before. Please explain what you mean - I understood it as conditioning on a random variable, with the given distribution.

- line 545 - I think there is a mistake in an index in the double sum.

- Fig. S1 - Why is the lapse rate 0 here, and not in the Brunton, et al paper? Are you regularizing? Are these parameters consistent with Hanks, et al 2015?

- I appreciated the model validation and simple example to explain the approach.

Reviewer #3 (Remarks to the Author):

This study examines how the frontal orienting fields in rats encode sensory evidence in a dynamic evidence accumulation task. The authors trained rats to perform a previously developed dynamic evidence accumulation task and quantified the provisional and final decisions using a previously-developed behavioral model. Using extracellular electrodes, the authors recorded from the FOF from

5 rats. The authors found that similar to their previous study, FOF neurons exhibited categorical coding for choice directions reflecting accumulated evidence. A new finding in this study is that FOF neurons tracks the dynamic changes in the provisional decisions accompanying the state changes in each trial. This is an interesting phenomenon, however, the message reached by this observation is vague. In previous studies from the same lab, much work has been done, including developing the evidence accumulation task and modeling the stationary or dynamic version of the task (Bruton et al, 2013; Piet et al 2018), and examining the neural activity and causal roles of PPC and FOF during the evidence accumulation task (Hanks et al, 2013; Erlich et al, 2015). In comparison, the overall progress made in this manuscript is quite limited. The major new observation made here is the activity of FOF showing selectivity changes correlated with state changes. But it is unclear whether FOF encodes motor preparation / motor planning that co-fluctuated with provisional decisions, or FOF is encoding the dynamic evidence accumulation (or provisional decision). An experiment of manipulating FOF during different time epochs along the dynamic accumulation process is missing. The role of FOF in the dynamic evidence accumulation task is still unclear.

Specific points

1. My first concern is that the title needs to be more precise with species, brain region and testing conditions. Rather than stating 'changes of mind' explicitly, which in my opinion was overly eye-catching than accurate, it would be helpful for the reader to know that the task has a dynamic nature which induces possible 'mind changes' as inferred by the experimenters, because there was a lack of direct behavioral evidence showing that rats indeed changed their "mind". 'Provisional or intended decision' might be more appropriate.

[1]
[SEP]

2. It is unclear whether FOF is playing a different role in the current version of task comparing to the previous stationary accumulation task, and also comparing to mouse ALM in the delayed response task. In the current manuscript, the authors stated that FOF neurons encoded evidence, and tracked the evidence changes throughout the dynamic accumulation process. However, this appear to be at odds with the conclusion reached in the Hanks et al, 2015 study, where it was stated that "The more categorical encoding found in the FOF suggests that, contrary to current views, this brain region may not be involved in the graded evidence accumulation process itself, but instead may be more involved in the conversion to a categorical choice." This is basically motor preparation. This view was further supported by FOF inactivation experiments in Hanks et al, 2015, showing that only peri-choice inactivation led to a significant ipsilateral bias, while inactivation during the early accumulation period produced no significant effect. In my view, the notion of encoding evidence accumulation should be distinguished from the notion of encoding movement planning. Although phenomenologically they are often difficult to separate, accurate perturbation experiment could provide important clues. Unfortunately, such manipulation experiment testing the causal role of FOF is missing in this manuscript. In addition, in Hanks et al, 2015, comparing the information coding in PPC and FOF suggested that PPC is more likely to encode accumulated evidence. It would be helpful to also examine PPC activity in this dynamic accumulation task, and compare the activity between these two regions.

3. Although this dynamic accumulation task is interesting in the sense of more closely simulating a volatile environment, this task also makes it more difficult to dissociate sensory evidence accumulation and motor planning. In this task, the choices were much more heavily dependent on the near-to-decision sensory information. According to Figure 1F, basically only the sensory information from the last 200 ms matters. This makes it difficult to distinguish whether a brain region is involved in sensory evidence accumulation or motor preparation, particularly for a region as FOF. The authors should explicitly discuss this and make more cautious interpretation of the information encoded in FOF neurons. Otherwise, the authors should provide additional analysis to distinguish whether FOF neurons were encoding movement signals or accumulated evidence, e.g., by looking at the FOF selectivity in error trials.

Response to Reviewers

Black, reviewer comments

Blue, our response

Purple, changes to manuscript text.

We thank the reviewers for their overall positive evaluation of our work, and the detailed feedback on our manuscript. We feel our updated manuscript has been greatly improved by their feedback, and we believe their concerns have been addressed. Broadly, the updated manuscript has improved mathematical notation, greater clarity on our analysis methods, additional supplementary figures, and improved discussion of the limitations and interpretations of our results. Below we respond to each item in detail. Note that line numbers mentioned by the reviewers may have changed. When we use a line number it refers to the updated manuscript.

In addition to the response to reviewers below, we have included data and code availability statements.

Data Availability

The datasets generated and analyzed in this study are available at https://figshare.com/articles/dataset/Manuscript_Data/16695592.

Code Availability

Analysis code used in this study is available at https://github.com/Brody-Lab/dynamic_ephys.

In addition to the reviewers' concerns below, we realized a minor coding error led to some trials being erroneously excluded from the PSTHs shown in Figure 2 and the definition of active cells. This coding error did not impact any other analyses, and did not substantially alter the PSTHs in Figure 2. Figure 2 has been regenerated with all available trials and the number of active cells is now reported as 579 instead of 592.

REVIEWER COMMENTS

Reviewer #1 (Remarks to the Author):

In this work, Boyd-Meredith et al. examine neural activity in the frontal orienting field (FOF), which has previously been shown to track accumulated evidence for a two-alternative auditory decision task. Here, they consider a non-stationary version of the same task that has been previously modeled and ask whether single neurons continue to reflect these decision variables in cases where evidence may not necessarily rise uniformly during the task. They argue that this is indeed the case, that deviations from mean firing across trials may be modeled as a separable function of time and accumulated evidence, and that the timing of population responses matches the timing of statistical inference as captured by the model.

There is a great deal to like about this paper. The presentation is clear; I found the analyses well described and results compellingly laid out. The work continues a line of research from these

investigators in an interesting and relevant direction and demonstrate the power of modeling tightly linked to neural physiology. I have some concerns, though, most related to presentation and nuancing results, but at least one serious.

Thank you for the positive evaluation of our work, we address each concern below.

Big picture things:

1. Despite some familiarity with the authors' previous work, I found the introduction a little narrowly focused and perhaps assuming a higher level of familiarity with recent FOF literature than a general reader might possess.

We thank the reviewer for pointing out where our manuscript was too narrowly focused. We have expanded the introduction to include references to the broader decision making literature, and a more careful introduction to the relevant FOF literature. We have also addressed issues raised by other reviewers, as explained below. The revised introduction now begins with the following two paragraphs (line 26):

When making decisions, animals must weigh and combine the available evidence in favor of each alternative. With each new observation, evidence about the underlying state of the environment gradually accumulates until the animal is ready to act. This accumulation model successfully describes a wide array of decisions (Gold, 2007; Krajbich, 2015; Ratcliff, 1978). Neural correlates of this accumulation process are also present across many brain regions in animals performing perceptual categorization tasks (Gold, 2007; Brody, 2016). Not all brain regions with neural correlates of evidence accumulation play the same role in the decision making process (Brody, 2016; Siegel, 2015; Katz, 2016). For example, regions important for accumulation may represent evidence in a continuous, graded fashion. On the other hand, regions important for reading out choice and preparing motor movements may have more categorical representations of the accumulated evidence.

Hanks et al (2015) characterized the neural representation of accumulating evidence in rats performing accumulation of trains of auditory click evidence. In the task, two streams of randomly-timed auditory clicks were emitted from either side of a fixation location and rats were trained to orient toward the side that played a greater number of clicks. Presenting the evidence as discrete pulses provided additional power to estimate the evolution of each subject's latent accumulated evidence variable on individual trials (Brunton, 2013), increasing the resolution for estimating neural encoding of this variable across brain regions (Hanks, 2015; Scott, 2017; Yartsev, 2018). Experimenters recorded from the posterior parietal cortex (PPC) and the frontal orienting fields (FOF), a frontal cortical structure implicated in short term memory and preparation of orienting movements (Kopec, 2015; Erlich, 2011; Ebbesen, 2018). They found that FOF neurons encoded the instantaneous accumulated evidence with sigmoidal tuning curves that remained stable during accumulation (Hanks, 2015). These representations were more categorical than representations found in PPC, providing a readout of the animal's provisional decision—the choice favored by the evidence presented so far—throughout

accumulation (Hanks, 2015; Yartsev, 2018). While this study could not differentiate between evidence representations resulting from a role in motor preparation and motor-independent evidence representations, temporally-precise perturbations of the signals in FOF only impaired the animal's choice when they overlapped with the final time points of accumulation and not when they occurred early in the evidence period. These results, along with a two-node model of the FOF (Piet, 2017), suggested that the FOF is not involved in the accumulation of new pieces of evidence, but provides a critical readout of the animal's provisional decision when it is time to act.

2. At several points, I would have appreciated more help with understanding possible alternatives to the authors' findings.

The reviewer raises an important consideration about possible alternative findings. Below we address this concern by documenting the assumptions and limitations of our data curation and analysis steps. These limitations inform what possible alternatives we could have found in our data.

For instance, given that the population of selected cells is chosen based on side responsiveness (essentially the difference between the means of the blue and green traces in Fig2A), it would have been good to know what possibilities this precludes. For instance, traces that oscillate or are u-shaped in time would presumably be missed by the selection criterion, which means that responses across trials in the surviving neurons should be either constant or roughly monotonic, correct?

The reviewer raises an important consideration about how our cell selection choices potentially impact our results and interpretation. In particular, the reviewer is concerned that we are excluding cells with firing rates that are u-shaped, oscillatory or otherwise non-monotonic over the course of the trial.

In fact, our selection criterion (significant difference in spike counts during the entire stimulus period for left choice versus right choice trial) ignores time. This means that included cells can have a variety of average firing rate trajectories over time. Oscillatory or u-shaped responses could be included as long there is sufficient modulation by choice at some point in the trial. For example, oscillatory firing rates might have a DC offset associated with side choice or more complicated effects like modulation of peak amplitude by upcoming choice. Similarly, u-shaped firing rates could pass the criterion as long as the magnitude of the u-shapes differs depending on upcoming choice. Example cell 17784 in Figure 2B may be an example of a u-shaped firing rate, which falls following trial initiation and then only rises at the end of the trial for right choices.

While the criterion doesn't place any obvious constraints on cells' tuning to time in the trial, it does constrain the cells' tuning to the accumulator value. We should expect the accumulation value tuning to be higher on one side of the decision boundary for at least part of the trial. This doesn't need to be true (or consistent) for the entire trial and the tuning doesn't need to be

monotonic. Cells could flip their side-preferences during the trial or be completely untuned for parts of the trial. Our cell selection criterion does exclude something like a u-shaped accumulation value tuning which is symmetrical around the decision boundary and pertains over the entire trial.

We have modified the text to clarify our selection criterion (line 176):

Cells were defined as pre-movement side-selective if their total spike counts during the trial between the start of the stimulus and the movement away from the fixation port were significantly different depending on the animal's side choice (2-tailed t-test, $p < .05$). This subset made up 17% of the active population ($n=103$ selective). For each neuron, the side associated with the higher spike count is referred to as the cell's preferred side. We focus on these pre-movement side-selective neurons because they are most likely to play a role in decision formation

In the results when we introduce the tuning curve analysis (line 202):

Here, we sought to test whether the FOF continues to stably encode the evidence throughout trials when the environment is dynamic. We reasoned that choice-encoding might emerge later in the dynamic environment when early provisional decisions are less likely to be acted upon. For example, neurons might only represent choice once the stimulus period has ended and the animal has committed to a decision. Further, we asked whether this encoding can still be captured by a single tuning curve in the dynamic environment. While our cell selection enforces that there be choice-encoding at some point in the trial, it does not constrain the presence or stability of this encoding over the course of the trial.

Likewise, for the rank-1 decomposition in (1), a separable rate function is clearly not the most general possibility, so what is the significance of the fact that this works? What would need to have been different for it to fail?

We thank the reviewer for this important question regarding the variance explained by the rank 1 decompositions. In order to fully address this concern, we reiterate some of the details of the process by which we compute the rank 1 decomposition. Before computing the rank 1 decomposition, we first construct the residual rate map $E[r|a,t]$ using the observed firing rates and inferred accumulation value distribution trajectories. We can think of this residual rate map as a set of tuning curves for each time bin, eg: $E[r|a,t=(0,25ms)]$ which reflect the neuron's true tuning to the accumulation value at each time point as well as corruption from trial-to-trial noise. These tuning curves describe the cell's firing rate modulation by the accumulation value for each point in time. Our rank 1 decomposition, $E[r|a,t] = f(a)*m(t)$, attempts to describe this set of tuning curves as a scaled modulation, $m(t)$, of some common tuning curve, $f(a)$.

For the rank 1 decomposition to succeed, the tuning curves at each time point must be linearly scalable versions of each other. Broadly, this can fail for two reasons. First, the cell's firing rate needs to be strongly modulated by the accumulation value relative to the trial-to-trial noise. If trial-to-trial noise dominates, the residual rate map will be highly variable at each timepoint. Second, the tuning curves need to be consistent at each time point. Tuning curves

can change sign, or be linearly scaled (which will be captured by $m(t)$), but their shape cannot change.

With this in mind, we have added an additional supplemental figure that examines where our rank 1 decompositions are successful and where they fail to capture the data.

First, not all cells are fully described by the rank 1 approximation. The average explained variance by rank 1 approximation across the selective population is high, suggesting the rank 1 approximation is a good description of many cells' firing rate tuning maps. However, there is a distribution over explained variance by the rank 1 approximation and it is instructive to look at the cells with lower explained variance. We have added a supplementary figure (Fig. S10), which contains the histogram of variance explained over the selective population and shows example tuning maps for cells at the bottom, middle and top of the range of variance explained.

Second, the degree to which the cells are described by the rank 1 approximation is correlated with the duration of the pre-choice side selectivity (Fig. S10C). This is consistent with the predicted relationship between consistent tuning throughout trials and the rank 1 explained variance. In the example cells (Fig. S10D-F), we see that the residual rate map of the cell with the least rank 1 explained variance appears to be dominated by noise early in the trial, rather than having a tuning curve that is changing in some non-linearly scalable manner.

We have added the following at line 231:

Higher explained variance indicates that a cell's residual rate map can be accurately described by a single tuning curve with linear scaling across time points. The fraction of the variance captured by the rank 1 decomposition is positively correlated with the total duration of side-selectivity favoring the cell's preferred side (Pearson's correlation, $\rho=0.41$, $p < .01$; Fig. S10C)

We have also added a supplementary figure titled Rank 1 decomposition for all side-selective cells (now Fig. S10).

Figure S10: Rank 1 decomposition for all side-selective cells. Comparison of variance explained by the rank 1 decomposition for side-selective cells. (A) Histogram of variance explained by rank 1 decomposition for all side-selective cells. Black vertical line marks the mean variance explained and purple arrows mark variance explained for the lowest, middle, and highest variance explained cells. (B) Variance explained as a function of the rank of the

decomposition for all cells (gray traces). Black trace marks the mean and error bars represent 95% confidence intervals. (C) Rank 1 variance explained as a function of the total duration of side-selectivity (significant AUC favoring for the cell's preferred side). These values are positively correlated (Pearson's correlation, $\rho=0.41$, $p < .01$). (D) Residual z-score firing rate map for example cells for the cell's with the least (left), most (right), and the middle (middle) variance explained by the rank 1 decomposition. (E) Rank 1 approximation for the cells in B with annotations showing the variance explained (VE). (F) Rank 5 approximations, plotted as in C.

Same thing for the analyses in Figure 4E and Figure 5. I *think* I understand the arguments here, but more discussion of how these relate back to the underlying scientific questions (especially with regard to possible alternatives) would be helpful.

We thank the reviewer for again raising important considerations about the limitations and possible alternative outcomes of our analyses. Both Figure 4 and 5 examine how FOF neurons encode evidence around the times of inferred changes of mind. Figure 4 examines changes in firing rate based on the inferred provisional choice. Figure 5 utilizes the behavioral model and tuning curve analysis to develop a finer picture of evidence encoding during changes of mind. In both analyses, we find that FOF neurons encode the provisional choice, even when that provisional choice differs from the final choice. These figures rule out 3 major possible alternatives.

First, the primary finding from Hanks et al, 2015, that FOF neurons encode the choice throughout the trial, did not have to generalize to a dynamic environment. In principle, the change in task demands could have led to a change in FOF representations during the task. For example, we might have seen that evidence encoding in FOF did not emerge until after the first 500 ms, the earliest possible response time. We might also have observed that evidence encoding in FOF did not emerge until after the final state change when the animal had fully committed to a decision.

Second, even though the tuning curve analysis aligned to stimulus onset in Figure 3 appears to show stable coding, this could have been an artifact of averaging over many different trials with different state change timing. For example, the tuning curve in Figure 3 could have been dominated by a subset of trials with no state changes, or strong coding. In this case, the change of mind-aligned analyses in Figure 4 and 5 would have been more prone to fail because they divide trials specifically into encoding before and after the model-predicted provisional choice changes. Examining neural activity surrounding model-predicted changes of mind can provide a more detailed interpretation of the result observed in Figure 3, particularly with respect to the timing of responses to changes of mind. We expand on this point in the response to the reviewer's next question.

Finally, the tuning curve analysis in Figure 5 strengthens the evidence for the effect observed in Figure 3 that the neural encoding of evidence is well described by a single tuning curve with a linear temporal modulation.

We have modified the manuscript in the following places to mention these possible alternatives.

To address whether the stationary environment results generalize to a dynamic environment (line 195):

To better describe encoding of the provisional choice throughout the trial, we applied and extended a method developed to quantify the tuning of single neurons to the accumulated value at each moment during the trial. Grouping firing rates according to the predicted accumulation values at each timepoint, allows us to more informatively combine information across trials with different hidden state change timing, final choice, and trial outcome (see Fig. S6 for population average of correct and error trials). Using this method, Hanks et al (2015) found that FOF neurons had a stable encoding of evidence throughout accumulation in a stationary environment. Here, we sought to test whether the FOF continues to encode the evidence throughout trials when the environment is dynamic. FOF cells could encode provisional choice throughout the trial, as observed in the stationary environment, or choice-encoding could emerge later in the dynamic environment. For example, neurons might only represent choice once the stimulus period has ended and the animal has committed to a decision

To address whether the results of Figure 3 could be driven by a subset of trials (line 253):

If cells are stably tuned to the accumulated evidence throughout deliberation, we should be able to see rapid responses in their firing rates to changes in the animal's provisional decision. Unlike the previous analysis (Fig. 3), here we isolate time points around inferred changes of mind, grouping data only by the provisional decision (the sign of the accumulation value). By examining neural responses before and after model-predicted changes of mind, we can ensure that the stable choice coding seen in Fig. 3 is not an artifact produced by averaging in a subset of trials with stronger coding and fewer changes of mind.

To justify the use of the accumulation model in figure 5 (line 294):

To further characterize cell tuning to accumulated evidence during changes of mind, we recomputed the tuning maps aligning time to model-predicted state changes instead of the start of the trial. This analysis is restricted to the time points around inferred changes of mind as in Fig. 4, but also allows us to more closely examine the stability of tuning before and after changes of mind.

We realized the title for Figure 5 was incorrect, reflecting an earlier analysis that was removed from the manuscript before the original submission. We apologize for this oversight, and we have retitled Figure 5 to better reflect the analysis presented

Stable tuning curve captures responses before and after state changes

Likewise II. 236-238: Given that firing rates are smoothed, and cells have been selected to have differencing firing rates by side, how can this fail? The assumption is that, if provisional decisions are reflected, then firing rates should transition between high and low as the evidence favors preferred versus non-preferred sides, correct? This may be what the authors are trying to say, but the point about an intermediate value needing to obtain between a high and low one seems to me trivial and perhaps not the clearest statement of the point.

We thank the reviewer for pointing out this point of confusion in our explanation of this effect. We agree that, given smoothing, it is trivial to observe intermediate values between a high and low value. However, we are interested in the timing of the intermediate value relative to inferred changes of mind. We observe that the timing of the intermediate value coincides with the model-predicted state change time (Fig. 4B-E). Notably, there is delay between the generative state changes and the timing of intermediate firing rates, indicating that the model-predicted state changes provide a better explanation of the firing rate changes than the generative state does. We have modified the text to clarify this important detail (line 281):

If cells are encoding the animal's provisional decision, we expect them to take intermediate firing rates during changes of mind and not show significant encoding of either state. If our behavioral model accurately predicts the timing of changes of mind, these intermediate firing rates should coincide with model-predicted state changes. As predicted, we find that the population reaches its minimum fraction of cells differentiating between states at the time of the model-predicted state change. We recomputed the timecourse of discriminability across cells triggered on changes in the veridical environmental state, rather than the model-predicted changes. When we do this, we find the time point at which the minimum fraction of cells significantly discriminates between states is delayed relative to generative state changes. This is consistent with the FOF tracking changes in the sign of accumulated evidence rather than simply responding to the instantaneous stimulus.

3. Notation: quibbles about notation are petty at best, but (1) the authors have generally been careful and precise, so I hope they will take this in that spirit; and (2) in a couple of cases, it did cause me real confusion.

We thank the reviewer for a careful reading of our notation. We have clarified our notation and believe we have improved the readability of the paper. We consider each issue below.

II. 543-547: Marginalization is an odd term here. Is this really, as written, $E[r|a,t] P(a, t)$? Why not use the conditional expectation? Similarly, the formula on l.545 is not $E[r|t]$ but $E[r|t]P(t)$. I would think that what the authors want is the conditional expectation (the average firing rate in each bin), but there seems to be some discrepancy between that and what they've written here. I'm unclear which they actually did.

On line 543, do they really mean $\sum_i r_i P(r_i|a, t)$? Similarly, in (2) and (18), is $E[r(t)]$ supposed to be $E[r|t]$?

We thank the reviewer for closely evaluating our equations and raising important clarifications on our methods. The reviewer is correct that the term marginalization and the mathematical expressions presented here do not refer to the appropriate operations. We have replaced usage of 'marginalization' with 'conditional expectation,' which is the correct term for what we did. Correspondingly, we have replaced all instances of $E[r(x)]$ with $E[r|x]$. The relevant passage of the methods section now reads (line 622) :

To estimate each cell's firing rate map, we compute the conditional expectation of the firing rate in each accumulation value and time bin $E[r|a, t] = \sum_r r P(r|a, t)$. We then computed the expected difference, at each time point, from the cell's time-averaged firing rate as a function of the accumulation value $E[\Delta r|a, t] = E[r|a, t] - E[r|t]$, where $E[r|t]$ is the average of $E[r|a, t]$ over all values of a . Average accumulation value tuning over the full trial is computed by averaging $E[\Delta r|a, t]$ over time to get $E[\Delta r|a]$.

Similarly in Figure 3, across the top, I would have expected to see $E[r|a, t]$, $E[r|a, t] - E[r|t]$, and $E[r|a] - E[r]$. It's not the Delta notation that I find problematic so much as the fact that it's unclear in the formulae which variables the expectation is being computed with respect to (plus my confusion about $E[r(a, t)]$ stated above).

We have replaced the titles in Figure 3 and 5 to reflect conditional expectations, consistent with the changes described above.

The authors mix notation throughout the paper, sometimes using E for expectation, sometimes $\langle \rangle$ and $\langle \rangle$ (in (4) and (5)), and it would help readers, I think, to be consistent.

We thank the reviewer for pointing out this inconsistency. We have changed equations 4 and 5 to use $E[\]$ for expectation instead of angle brackets. We have also changed the variable names to remove repurposing of variable names used elsewhere. Specifically, in equations 4 and 5, the excess click rate was defined as $r(t)$ and we have changed it to $e(t)$. We hope this further clarifies the text.

4. I have two confusions here that may either be misunderstandings or serious concerns:

4.1: On ll. 437-440, it is stated that "the forward and backward distributions are conditionally independent, conditioned on the final value of the accumulated evidence." I *think* what the authors mean here is stated mathematically in (15). However, unless I am mistaken, the past and future distributions of the variable a are not independent unless one is also given the value of C . That is, unless one knows the value of $C(t)$, the process is no longer Markov, since one is missing the filtered information about the past contained in C .

We thank the reviewer for raising an important question about our analytical calculation of the posterior distribution. The reviewer is correct that the forward and backward distributions would not be independent if the adaptation value C was a stochastic process. However, adaptation is modeled as a deterministic process, consistent with Brunton et al (2013) and Hanks et al (2015). That is, the adaptation effects of consecutive clicks are applied prior to the corruption of clicks by per-click noise. This can be thought of as adaptation at the sensory level, upstream of the accumulation process. As a result, it is easy to use the known value of $C(t)$ in the backward distribution independent of the noise realizations in the forward model. Below, the reviewer notes that we did not include the equations for C , which would have helped clarify this question. We apologize for the lack of clarity. The relevant equations are now included in the text, as

described below. To help clarify this issue in the manuscript we have added the following section to the methods section when discussing the computation of the posterior distribution (line 807):

It is important to note here that the adaptation process C is deterministic, and its evolution does not depend on the stochastic per-click noise realizations. We can consider this as an upstream sensory adaptation that happens before the integration process. As a result, the adapted clicks can be included in the forward and backward distributions.

4.2: In (14), the authors condition on y to produce the backward posterior $b(a)$. However, instead of conditioning on the set $[a_N \in [B, \infty))$ for $y=1$, they additionally assume a uniform marginal distribution on the set. I have no idea why this should be true. More correctly, it can't be --- not just because the uniform distribution here is not normalizable, but because the marginal distribution of their stochastic process with Poisson increments should be Gaussian with a variance set by (24) and (25). Indeed, one should calculate the backward distribution by starting with the correct marginal for a_N , which is the approach originally used in Brunton, Botvinick, and Brody (for a discretized a). It is not at all to me clear how (28) and (29) approximate this unless the w_i are *not* all the same but are instead the marginal probabilities of a_N being in the given mixture component. I am not sure whether this affects the results of the paper, but this approach seems to me incorrect.

The reviewer raises important concerns about our calculations of the posterior distribution. We believe the underlying issue here is a lack of clarity on our part about what we are trying to compute (the posterior distribution), and how we are computing it (an intermediate backward distribution). This lack of clarity is compounded by different terminology used in Brunton et al (2013). We have modified the manuscript in several ways to address this confusion, and we go through our reasoning step by step below. The main modifications are (1) clarifying the end goal is the computation of the posterior distribution, (2) clarifying we compute the posterior distribution as the product of a forward and backward distribution, (3) our backward distribution is not the same as the backward pass distribution in Brunton et al (2013), indeed what that paper called the backward pass distribution is what we refer to as the posterior distribution.

Whereas Brunton et al (2013) used the terms backward pass distribution to describe the posterior distribution, we use the term backward distribution to refer to something distinct from the posterior distribution. This repurposing of terminology will be confusing to readers who are very familiar with the Brunton work. We have reworded these descriptions and attempted to highlight the ways in which our approach differs from the approach in Brunton et al (2013). Moreover, we now refer to the posterior as $p(a)$ to more clearly differentiate it from the forward distribution $f(a)$ and the backward distribution $b(a)$.

Having clarified terminology, we now address the reviewer's central concern about the validity of our computation of the posterior distribution. The reviewer is correct that the *posterior* distribution should not be uniformly distributed. The important clarification is that equations 14, 28, and 29 describe the backward distribution $b(a) = P(a|t, \delta_R, \delta_L, \theta, y)$, not the posterior

distribution $p(a) = P(a|t, \delta_R, \delta_L, \theta, a_0 \sim N(0, \sigma_i^2), y)$. The backward distribution makes no assumptions about the initial distribution of accumulation values, but instead assumes that the final accumulation value is consistent with the rat's choice, y . Taking the product of the forward and backward distributions combines these assumptions to produce the posterior $p(a) \propto f(a)b(a)$. Multiplying $b(a)$ by $f(a)$ introduces the appropriate weighting into each mixture component of the posterior distribution.

We reworded the description of the forward, backward and posterior distributions in the methods as follows (line 492):

We develop a novel method of computing the posterior distribution by taking the product of the forward distribution and a backward distribution. Again, we note that while Brunton et al (2013) refers to the posterior distribution as the backward pass distribution, we use the term backward distribution to refer to a distinct distribution which constrains the final state of the accumulation value distribution in accordance with the animal's choice, but does not constrain the initial state. As described above, the forward distribution assumes that the initial accumulation value a_0 is normally distributed with mean 0 and variance σ_i^2 . The backward distribution makes no assumption about the initial distribution, but assumes that the final accumulation value a_N is uniformly distributed on the side of the decision boundary B that corresponds to the rat's choice. Importantly, the forward and backward distributions are conditionally independent, conditioned on the final value of the accumulated evidence. Given that these distributions are independent, their product gives the posterior distribution $p(a)$ that combines the constraints on the initial and final distributions of accumulation values

$$p(a) = P(a|t, \delta_R, \delta_L, \theta, a_0 \sim N(0, \sigma_i^2), y) \quad (16)$$

$$\propto f(a)b(a) \quad (17)$$

Where $f(a)$ is the forward model described above, which assumes a Gaussian initial distribution of accumulation values depending on σ_i^2 :

$$f(a) = P(a | t, \delta_R, \delta_L, \theta, a_0 \sim N(0, \sigma_i^2)) \quad (18)$$

and $b(a)$ is the backward distribution, which assumes the final distribution of accumulation values is uniformly distributed over the side of the decision boundary B corresponding to the animal's choice y :

$$b(a) = P(a | t, \delta_R, \delta_L, \theta, y) \quad (19)$$

$$\begin{aligned}
&= P(a \mid t, \delta_R, \delta_L, \theta, a_N \mid U(B, \infty)) && \text{if } y = 1 \\
&= P(a \mid t, \delta_R, \delta_L, \theta, a_N \mid U(-\infty, B)) && \text{if } y = -1.
\end{aligned}$$

(20)

Note that we have reordered and expanded the equations so that the posterior distribution appears in the text before the backward distribution. We hope that this clarifies the motivation for computing this separate backward distribution as an intermediate step in computing the posterior distribution.

In the supplemental methods, we have removed the redundant descriptions of the forward model and included similar clarifying statements differentiating the posterior distribution from the backward distribution, which we will not repeat here. These changes begin on line 792.

We hope that this clarifies the confusions in the text. Regarding the soundness of our approach, we note that our posterior model validation section shows that our approximations produce the appropriate results for the posterior distribution (Supplementary Figures 6-8).

Minor points:

1. I. 47: This may not mean much to those less familiar with the FOF literature.

The reviewer points out that our description of previous FOF results was too brief. We have reworded the last 4 sentences of this paragraph in hopes of making the findings clearer (line 44):

They found that FOF neurons encoded the instantaneous accumulated evidence with sigmoidal tuning curves that remained stable during accumulation (Hanks, 2015). These representations were more categorical than representations found in PPC, providing a readout of the animal's provisional decision—the choice favored by the evidence presented so far—throughout accumulation (Hanks, 2015; Yartsev, 2018). While this study could not differentiate between evidence representations resulting from a role in motor preparation and motor-independent evidence representations, temporally-precise perturbations of the signals in FOF only impaired the animal's choice when they overlapped with the final time points of accumulation and not when they occurred early in the evidence period. These results, along with a two-node model of the FOF (Piet, 2017), suggested that the FOF is not involved in the accumulation of new pieces of evidence, but provides a critical readout of the animal's provisional decision when it is time to act.

2. II. 62-63: This presumes that the animals are performing the hypothesized strategy. It may be cleaner to simply talk about static vs. dynamic environments and separate what is experimenter-controlled and what is results.

The reviewer raises an important consideration that the experimental control of the static vs dynamic environment is a more direct observation than the inferred strategy used by the animals. While we feel confident in our evaluation of the rat's strategies based on previous studies (Brunton et al (2013), Piet et al (2018)), we have modified the text to address the reviewer's concern. It now reads (line 67):

However, it is unknown whether the neural correlates of evidence accumulation observed during putatively non-leaky integration in stationary environments are preserved in animals performing putatively leaky integration in dynamic environments.

3. I realize the answer is in previous papers, but the equation governing C is never given.

We apologize for this oversight. We realized we also never gave the differential equations for the accumulation model, only the equations for its solution. We now include the equations for the accumulation model, and the adaptation variable C. We added the following to the methods section (line 453):

The accumulation model is a stochastic differential equation that describes the evolution of an accumulation value a , and a sensory adaptation value C :

$$da = (\delta_R \eta_R C - \delta_L \eta_L C)dt - \lambda a dt + \sigma_a dW, \quad (6)$$

$$dC/dt = (1 - C)/\tau_\phi + (\phi - 1)C(\delta_R + \delta_L). \quad (7)$$

Each sensory click is scaled by the sensory adaptation value C and multiplicative gaussian noise η which is drawn from $N(1, \sigma_s)$. The model parameters θ can be described in words as an initial noise variance σ_i^2 , a per-click noise variance σ_s^2 , a memory noise variance σ_a^2 , a discounting rate λ , the strength and time constant of adaptation ϕ and τ_ϕ , a decision boundary B , which captures the animal's bias, and a lapse rate l . If the adaptation strength parameter $\phi < 1$ then consecutive clicks are depressed. If $\phi > 1$, then consecutive clicks are facilitated.

The forward model is the solution to Eqn. 6, assuming the initial accumulation value a_0 is

Gaussian distributed with zero mean and variance σ_i^2 .

Reviewer #2 (Remarks to the Author):

This manuscript can be viewed as a followup to the study of Hanks, et al 2015 (reference 6). Using similar methods the authors show that the activity of neurons in FOF represent provisional decisions in their firing rates even when changes in the environment lead to a change of mind.

The manuscript is interesting, and well written. The analysis is largely based on that in previous work by this group, but offers several extensions, and confirms previous findings in a more ethologically relevant setting. I believe the results and the approach are solid. I have some comments on the presentation, and clarifying questions the answers to which may help a general reader as well.

We thank the reviewer for the positive evaluation of our work and feedback. We address each point in detail below.

- Figure 3 is the highlight of the paper, but I found it confusing when returning to it after reading further. First, this demonstrates that the responses in FOF are categorical. However, the first and last column show graded responses. This argument is clear in light of the Hanks, et al 2015 paper, where FOF responses are compared to those of different areas. However, it should be clarified here.

The reviewer raises an important concern about how to interpret the tuning curves presented in Figure 3. Hanks et al 2015 made a claim about FOF tuning being categorical by comparing FOF tuning curves to PPC tuning curves. We changed the introduction to clarify that previous descriptions of FOF responses as categorical were in contrast to other brain regions showing more graded tuning (line 44):

They found that FOF neurons encoded the instantaneous accumulated evidence with sigmoidal tuning curves that remained stable during accumulation (Hanks, 2015). These representations were more categorical than representations found in PPC, providing a readout of the animal's provisional decision—the choice favored by the evidence presented so far—throughout accumulation (Hanks, 2015; Yartsev, 2018).

In the current study we are not comparing FOF units with other brain regions, and further our central claim is not about the degree of categorical encoding (the slope of the tuning curves in figure 3), but rather if and how those tuning curves change around the time points of inferred changes of mind. Thus in the current study the central claim of Figure 3 is not that the tuning curves are more or less categorical than another brain region, but that cells stably encode evidence throughout the trial despite dynamic trials. This logic is outlined starting on line 193:

This creates frequent dissociations during the trial between the animal's provisional choice and the final choice, which are rare in stationary environments. To better describe encoding of the provisional choice throughout the trial, we applied and extended a method developed to quantify the tuning of single neurons to the accumulated value at each moment during the trial. Grouping firing rates according to the predicted accumulation values at each timepoint, allows us to more informatively combine information across trials with different hidden state change timing, final choice, and trial outcome (see Fig. S6 for population average of correct and error trials). Using this method, Hanks et al (2015) found that FOF neurons had a stable encoding of evidence throughout accumulation in a stationary environment. Here, we sought to test whether the FOF continues to encode the evidence throughout trials when the environment is dynamic. Further,

we asked whether this encoding can still be captured by a single tuning curve in the dynamic environment. Our cell selection enforces that there be choice-encoding at some point in the trial, but does not constrain the stability of this encoding over the course of the trial.

We have changed the title of Figure 3 to:

FOF neurons encode the accumulated evidence throughout the trial despite a changing environment

What confused me further is whether this was data from the entire trial and thus averaged over changes in environment? This should be explained clearly - and contrasted to how the data was analyzed in Figs. 4 and 5.

The reviewer raises an important clarification about what data goes into Figures 3, 4, and 5. The reviewer is correct that the data in Figure 3 is from the entire trial and thus the analysis averages together trials where the hidden state changes at different times. We apologize for not making this clearer, as it is a key point for the reader to understand.

In Figure 2, we see that FOF units encode the final choice of the animal. In Figure 3 we ask whether FOF units encode the accumulation value throughout the trial, despite dynamic changes in the environment. This analysis separates firing rates as a function of the accumulation value at each timepoint, but each bin averages together trials where the hidden state changes at different times. In principle, the tuning observed at each timepoint could be driven by subsets of trials in which that rat has already committed to its final decision at that timepoint. The analyses in Figures 4 and 5 rule out this possibility by realigning trials to state changes and showing that provisional choice (Figure 4) and accumulation value (Figure 5) are encoded as expected not just after state changes, but before them as well. We have clarified this in the text in the following places.

In the introduction to Figure 3, Line 193

This creates frequent dissociations during the trial between the animal's provisional choice and the final choice, which are rare in stationary environments. To better describe encoding of the provisional choice throughout the trial, we applied and extended a method developed to quantify the tuning of single neurons to the accumulated value at each moment during the trial. Grouping firing rates according to the predicted accumulation values at each timepoint, allows us to more informatively combine information across trials with different hidden state change timing, final choice, and trial outcome (see Fig. S6 for population average of correct and error trials).

In the introduction to Figure 4, Line 255

Unlike the previous analysis (Fig. 3), here we isolate time points around inferred changes of mind, grouping data only by the provisional decision (the sign of the accumulation value). By examining neural responses before and after model-predicted changes of mind, we can ensure that the stable choice coding seen in Fig. 3 is not an artifact produced by averaging in a subset of trials with stronger coding and fewer changes of mind.

In the introduction to Figure 5, Line 296

This analysis is restricted to the time points around inferred changes of mind like Fig. 4, but also allows us to more closely examine the stability of tuning before and after changes of mind.

- The description of the behavior model starting on line 99 could be a clearer. For instance, what does “this distribution” on line 103 refer to? The authors use a discrete time model, but the task occurs in continuous time. A lot of this is clear if a reader is familiar with the authors’ previous work. However, a novice reader will be confused.

We thank the reviewer for pointing out confusing sections of our model description, we have made several changes outlined below.

The reviewer is correct that on line 103 we have not yet properly defined the distribution over accumulation values. We have modified the order of presentation as follows (now line 110):

The model parameterizes the process by which the evidence available in each auditory click is integrated over time into a decision variable that guides the rat's choice.

The decision variable, referred to as the accumulation value a , takes an initial value a_0 , drawn from a Gaussian with zero mean and an initial variance σ_i^2 , which is fixed across trials.

Regarding discrete versus continuous time: since we have an analytical expression for the mean and variance as a function of time (equations 9 and 10), our model fitting procedure and predictions operate in continuous time. However, the reviewer is correct that for the firing rate maps where we relate firing rates to the posterior distribution $p(a)$, we discretize time. We add the following clarification to the methods (line 614):

When estimating the joint distribution we discretized our data along three dimensions: time, firing rate, and accumulation value. Time was binned into 25ms bins. Each neuron's firing rate was divided into 100 bins spanning the minimum to the maximum firing rate of the neuron. The firing rate bin was chosen by taking the average firing rate within the time bin. Accumulation value bin size was divided into 10 bins with width set to 1.625 except the last bin which was larger to capture the tails of the distribution. The posterior distribution was evaluated with 1ms time bins and accumulation value bin size of 0.1 and then downsampled to populate the joint distribution.

- Following on the previous remark, the authors need to carefully check notation. First, some variables are overloaded: r stands for rate in the main text, but also for the filtered clicktrain in the methods. B stands for bias and decision boundary and bias. Most importantly, some quantities sometimes come with subscripts, and sometimes do not - for example a_0 and a . The subscripts are time bin indexes for some variables, but denote neuron number elsewhere. Some

indices are not defined - eg I assume that a_N denotes the value of a in the last bin, N . However, I did not see N defined. A reader should not have to figure this out, nor have to go to wprevious papers to do so.

- A related point: Is B really a decision bound? The methods only mention the bias. I think they are equivalent in this context, but it may be best to keep the nomenclature consistent.

We thank the reviewer for pointing out these issues with the variables. We agree that readers should not have to guess about the meaning of subscripts and variable names, nor should they have to deal with egregious repurposing of variable names and subscript meanings. We have done our best to limit repurposing of names and subscripts and to make these very explicit when they are unavoidable. Below we outline the changes to our notation, and hope they clarify our model and methods.

We now use the variable d to describe the filtered click difference rates instead of r when describing the psychophysical reverse correlations (Eqns 3, 4).

We also clarify the notation for the click trains by adding the following (line 430):

Here, the click train δ_R is a sum of delta functions with peaks at the time of each right click and the value 0 everywhere else.

In the accumulation model, subscripts are used to refer to the initial and final values of a . We now state explicitly that a_0 is the initial value of a and a_N is the final value of a . The text was modified to read (line 114):

The decision variable, referred to as the accumulation value a , takes an initial value a_0 , drawn from a gaussian with zero mean and an initial variance σ_i^2 , which is fixed across trials.

And on line 126

At the end of non-lapse trials, the rat chooses to go right if the final accumulation value a_N is greater than the decision boundary B , and chooses to go left if $a_N < B$.

We no longer use subscripts to refer to trial number, or neuron number. We now only refer to trial number in equation 15, using superscripts.

We still use the subscript i to indicate the initial noise variance σ_i^2 , but add the clarification that this parameter is fixed across trials (line 113).

The decision variable, referred to as the accumulation value a , takes an initial value a_0 , drawn from a gaussian with zero mean and an initial variance σ_i^2 , which is fixed across trials.

In describing the grid solution for computing the backward distribution, we now index the bins with the variable j (instead of i) to reduce any possible confusion about this subscript.

We apologize for the confusion about the decision boundary parameter B . We have been inconsistent in referring to this parameter as both “bias” and “decision boundary”. In our model, the rat chooses to go right if the final value of a is greater than B , and goes left if the final value of a is less than B . We call this a decision boundary, a term often used to describe the hyperplane used by a classifier to separate class labels. The ideal value of B is 0, any deviation from 0 reflects a bias in the animal's choices. Therefore, the decision boundary encodes the animals' left or right bias. Importantly, this decision boundary is not the same as decision bounds used in other studies, which typically set a maximum magnitude that the decision variable can reach and in some cases set a threshold at which the animal commits to a decision and stops integrating new information. We have revised the text to always refer to this parameter as the decision boundary parameter. These are important distinctions and we attempt to clarify in the text.

On Line 126:

At the end of non-lapse trials, the rat chooses to go right if the final accumulation value a_N is greater than the decision boundary B , and chooses to go left if $a_N < B$. The ideal value of B is 0 and any deviation from 0 reflects the animal's side bias. On a fraction of trials l , called “lapse” trials, the rat chooses randomly (Fig. 1C). Unlike the model described by Brunton et al. (2013), there is no decision bound setting the maximum magnitude of a at which the animal is fully committed to a decision. Instead, the decision remains sensitive to new information throughout the accumulation period. Best fit values of this parameter were previously found to be effectively infinite and did not improve model fits (Brunton, 2013).

- A more general question - The authors assume that the rats use a normative model. This is consistent with Piet, et al. However, can simpler models be eliminated? For instance, the model where the rat goes with the side where it heard the last click or pair of clicks?

The reviewer raises important questions about the strategy the rats are using, and whether our methods make limiting assumptions about those strategies. While it is impossible to rule out all possible alternative strategies, our evidence accumulation model is highly flexible and our model fits are consistent with behavioral performance measurements. As a result we describe the rat behavior using a model-fit, not an explicit assumption of normative behavior.

In more detail, because of the model's flexibility, it can capture non-normative strategies. For example, the model can describe the last click or last pair of clicks strategy by increasing the level of discounting. When λ is more negative, later clicks are more heavily favored.

We also present the psychophysical reverse correlations, which provide a model-independent way of measuring the effect of clicks on choices throughout the trial (Fig. 1F). Additionally, a last

click strategy would not predict the improvement in performance associated with increased final state duration (Fig. 2E). These measures provide an independent corroboration of the model estimate of the discounting rate. For additional discussion of ruling out last-click strategies see Supplementary Note 10 in Piet et al 2018.

Finally, the fact that neural tuning tracks the model-predicted provisional decision supports the idea that the rats are using a strategy similar to that described by the model.

It is nevertheless worth noting that, while our model captures behavior on average, there are undoubtedly variables that influence behavior that we do not account for. For example, we do not model the effect of previous trial outcomes or serial side biases. We also do not try to measure fluctuations around the best fit parameters within or across sessions, which is beyond the scope of this paper, but could improve the description of the behavior.

We have modified the text to mention these important issues in several places. When we introduce the behavioral model in the results (line 135):

This model is highly flexible and can capture many possible behavioral strategies (Brunton, 2013; Piet, 2018).

At the end of the results section describing the model fitting (line 152):

The best fit model parameters, along with the model-independent behavioral assays described above, provide converging lines of evidence that the rats integrated evidence throughout the trial, with hundreds of milliseconds and multiple sensory clicks influencing their final decision.

In the discussion (line 356):

One important limitation of our study is that our evidence accumulation model only uses one fixed set of parameters to describe each rat's behavior in a trial in terms of the stimulus on that trial. While the model is highly flexible and captures average behavior, it does not allow parameters to change over trials, nor does it capture trial-to-trial history effects. Future work should develop more flexible behavioral models to capture slow drifts and sudden state changes in the parameters that describe the animals' strategies. This work will allow deeper investigation into neural coding.

- Fig. 2A - It would be informative to see the variability in the responses, represented as an envelope, or by a sample of traces shaded more lightly overlaid over the average. A supplementary figures is fine.

We thank the reviewer for suggesting improving the PSTH plots by showing the variability in responses. We have modified the main figure to show the s.e.m. in shaded regions around the main trace for the example cells. We also add two supplementary figures. The first (Fig. S5) shows trial to trial variability by showing spike raster plots for a random subset of 200 trials for each of the example cells. The second (Fig. S6) shows the population average responses conditioned on choice and outcome with the s.e.m. in shaded regions.

We have modified the text to point the reader to these supplemental figures (line 164):
Individual cells show stereotyped temporal dynamics aligned to both the onset of the trial (entering the center nose port), and the movement following the end of the stimulus (nose out of center port) (Fig. 2B; see Fig. S5 for spike rasters)

(line 188):

We observe divergences at different latencies depending on the final state duration (see Fig. S6 for population average conditioned on side-choice and trial outcome as in Fig. 2B).

Figure S5: **Example cell rasters** Trials are sorted by side-choice and stimulus duration. Trials are randomly subsampled to include 200 trials for each cell. Spikes are colored by side-choice (blue for left choices; green for right choices). Red points mark the end (left) or beginning (right) of the stimulus period.

Figure S6: **Selective cell population average PSTH** Average activity of all pre-movement side-selective cells ($n=103$) conditioned on side choice, outcome and cell preference. Purple traces represent choices of the cells' preferred side and yellow traces represent choices of the cells' non-preferred side. Dark, solid traces represent correct choices and light, dashed traces represent errors. Shaded regions represent s.e.m.

- "Recordings from 69 sessions yielded 738 units." - for all animals?

We have clarified that our units came from 5 animals (line 160)

Recordings from 69 sessions yielded 738 units across 5 animals.

- Please explain the AUC computation better, either in the methods or supplement. The other techniques are explained well (modulo issues with notation described above), so it will be good to have a self-contained manuscript.

We have added the following to the methods section, including a reference to Hanley & McNeil, 1982 (line 595):

In this application, the receiver operating characteristic curve (ROC) treats the spike rate in a time bin as a classifier of right versus left choice, computing the true positive rate and false positive rate as a function of the spike threshold. The area under this curve is equivalent to the probability that in a randomly chosen pair of right left choice trials the firing rate in that bin will be higher on the right choice trial than on the left choice trial. AUC values significantly greater than 0.5 indicate a preference for right choice trials and AUC values significantly less than 0.5 indicate a preference for left choice trials.

- The paragraph starting on line 176 is central, but it is a bit tough to follow. This is partly due to the notational issues mentioned above. however, I suggest carefully rewriting this paragraph to improve clarity.

We have attempted to improve the clarity of our notation. We have also corrected an error present in this section where we incorrectly described computing the conditional expectation as marginalizing. Finally, we have further reworded the paragraph for clarity, as follows (now line 211):

We used the approach described by Hanks et al (2015), to produce a map describing each neuron's tuning to the accumulation value over the course of the trial. First, we computed the joint distribution $P(r, a, t)$ of each cell's firing rate r , the instantaneous accumulation value a , and time in the trial t . The evolution of the distribution over a in response to right and left click trains δ_R and δ_L , given by the behavioral model described above, was further constrained using the animal's choice y on each trial, giving the posterior distribution

$p(a) = P(a|t, \delta_R, \delta_L, \theta, a_0 \sim N(0, \sigma_i), y)$. We improve on the method used by Hanks et al., (2015) by using an analytical computation of the posterior distribution of accumulated evidence, allowing for more accurate estimation of $P(r, a, t)$ (see methods). Firing rate maps are generated by computing the conditional expectation of the firing rate as a function of accumulated evidence and time, for each cell $E[r|a, t]$. We present this rate map for an example cell which is strongly tuned to the accumulator throughout the trial, firing more when accumulated evidence favors left choices (Fig. 3A). Because our neurons have stereotyped temporal dynamics aligned to stimulus onset, we subtract out the average temporal dynamics to get the expected firing rate modulation $E[\Delta r|a, t] = E[r|a, t] - E[r|t]$ (Fig. 3B). Following Hanks et al., (2015), a summary tuning curve was computed by averaging over time to get $E[r|a]$ (Fig. 3C).

- line 220 - decision boundary us defined as $a = 0$. What about bias?

The reviewer is correct that we should have used timepoints where the posterior mean accumulation value crosses the rat's best fit decision boundary parameter ($a=B$) to determine the timing of changes of mind rather than when it crosses the ideal observer's decision bias ($a=0$). We have recomputed all state change triggered analyses and correspondingly regenerated Figures 4 and 5. Since the decision boundary parameter is very small (reflecting small biases), this only affects the timing of changes of mind by a few milliseconds and produces a nearly indistinguishable set of Figures. We have corrected this line in the text to reflect that we use decision boundary B to determine the timing of changes of mind. We have also corrected a line in the methods, which now reads (line 646):

We defined model-predicted state changes as time points where the running average of the mean of the posterior accumulator value crossed the rat's best fit decision boundary B .

- eq (3) - $r_{i,j}$, notation overloaded

We have reworked the notation as described above, replacing r with d and dropping the subscript i , which is not necessary here.

- eq 13 and 14 - conditioning on $a_0 = \dots$ and $a_N = \dots$. I have not seen this notation before. Please explain what you mean - I understood it as conditioning on a random variable, with the given distribution.

We now introduce a_0 and a_N as the initial and final value of a in the results section (line 113) .

The decision variable, referred to as the accumulation value a , takes an initial value a_0 , drawn from a Gaussian with zero mean and an initial variance σ_i^2 , which is fixed across trials.

We repeat this phrasing in the description of the forward model in the methods to avoid confusion.

We are also now using the notation $a_0 \sim N(0, \sigma_i)$ to more clearly indicate that the value is drawn from a distribution rather than equivalent to the full distribution. Similarly, we use the notation $a_N \sim U(B, \infty)$ to indicate that when used in the backward distribution our knowledge of the final accumulation value is only constrained by the choice.

- line 545 - I think there is a mistake in an index in the double sum.

Thanks for catching this error. We have removed this equation, and instead described what we did in words. The paragraph has also changed due to our correction of the terminology and notation for taking conditional expectations. The paragraph now reads (starting line 622):

To estimate each cell's firing rate map, we compute the conditional expectation of the firing rate in each accumulation value and time bin $E[r|a, t] = \sum_r rP(r|a, t)$. We then computed the expected difference, at each time point, from the cell's time-averaged firing rate as a function of the accumulation value $E[\Delta r|a, t] = E[r|a, t] - E[r|t]$, where $E[r|t]$ is the average of $E[r|a, t]$ over all values of a . Average accumulation value tuning over the full trial is computed by averaging $E[\Delta r|a, t]$ over time to get $E[\Delta r|a, t]$. The same procedure is used to compute a map of z-scored firing rates. And these maps are averaged across pre-movement side-selective cells to produce a population average.

- Fig. S1 - Why is the lapse rate 0 here, and not in the Brunton, et al paper? Are you regularizing? Are these parameters consistent with Hanks, et al 2015?

The reviewer raises an interesting question as to why the lapse rate parameter for rats in a dynamic environment are 0 (this study and Piet et al, 2018), whereas in stationary environments they tend to be small but non-zero (Brunton et al 2012, Hanks et al 2015). We are not

regularizing the lapse rate parameter in this or previous studies. The rats presented here have parameters consistent with the rats in Piet et al 2018 (10 of 14 rats with lapse rate = 0). There are several possible explanations for this difference in parameters across stationary versus dynamic environments. First, rats may behave differently such that “random guessing” or other behaviors that would show up in the lapse rate parameter may be used less in a dynamic environment. Second, in dynamic environments our model does not include “sticky” evidence bounds. In the model, the sticky boundaries absorb probability mass that gets to certain +/- evidence levels. These bounds are not necessary when the rats use large evidence discounting rates (λ). It is possible that the sticky boundaries create the need for lapse rate parameters in the model.

We have added the following to the caption of Figure S1.

Brunton et al (2013) also included sticky (absorbing) evidence boundaries as an additional parameter. Consistent with Piet et al (2018), we did not include this absorbing parameter because it was previously found not to improve model fits (Brunton et al., 2013) and because the large discounting rate λ prevents the boundaries from influencing the model. The difference in lapse parameters may be explained by the large evidence discounting rates λ , the lack of absorbing bounds in the model, or simply a difference in rat behavior.

- I appreciated the model validation and simple example to explain the approach.

Thank you.

Reviewer #3 (Remarks to the Author):

This study examines how the frontal orienting fields in rats encode sensory evidence in a dynamic evidence accumulation task. The authors trained rats to perform a previously developed dynamic evidence accumulation task and quantified the provisional and final decisions using a previously-developed behavioral model. Using extracellular electrodes, the authors recorded from the FOF from 5 rats. The authors found that similar to their previous study, FOF neurons exhibited categorical coding for choice directions reflecting accumulated evidence. A new finding in this study is that FOF neurons tracks the dynamic changes in the provisional decisions accompanying the state changes in each trial.

This is an interesting phenomenon, however, the message reached by this observation is vague. In previous studies from the same lab, much work has been done, including developing the evidence accumulation task and modeling the stationary or dynamic version of the task (Brunton et al, 2013; Piet et al 2018), and examining the neural activity and causal roles of PPC and FOF during the evidence accumulation task (Hanks et al, 2013; Erlich et al, 2015). In comparison, the overall progress made in this manuscript is quite limited. The major new observation made here is the activity of FOF showing selectivity changes correlated with state changes. But it is unclear whether FOF encodes motor preparation / motor planning that co-fluctuated with provisional decisions, or FOF is encoding the dynamic evidence accumulation (or provisional decision). An

experiment of manipulating FOF during different time epochs along the dynamic accumulation process is missing. The role of FOF in the dynamic evidence accumulation task is still unclear.

The reviewer raises important considerations about the interpretation of our work, and how it fits into the existing literature. We thank the reviewer for raising these concerns and we have rewritten several sections of the text to clarify our interpretation of the findings. Before detailing the changes to the text, we provide a direct response to the reviewer's concerns to clarify the claims that we intended to make in the text.

In particular, the reviewer is concerned that our study cannot determine whether FOF responses encode "motor preparation" or "the dynamic evidence accumulation (or provisional decision)". We agree that our study cannot separate these potential drivers of FOF activity. It seems that at least part of the confusion stems from our use of the term "provisional decision". In our usage, the term "provisional decision" refers to the side-choice favored by the animal's decision variable at any given movement during the accumulation process. As in Hanks et al (2015), an encoding of the provisional decision provides an answer to the question "if the go signal came now, which choice should I make?" Importantly, this signal does not imply a role in the evidence accumulation process itself. Nevertheless, it remains unclear whether the observed signal is best described as a motor preparatory signal or as an abstract, motor-independent categorization of the accumulated evidence.

Hanks et al (2015) also did not attempt to fully separate motor encodings and evidence encodings. In that study, as is here, the sign of the evidence is perfectly correlated with the rewarded motor action. In our view, to fully distinguish between these possibilities, we would need to train rats to accumulate evidence agnostic of the response contingencies, which is out of the scope of the present work. However, Hanks et al (2015) did perform an additional optogenetic manipulation, which showed that FOF inactivations did not affect choice unless they occurred immediately prior to the response period. This, along with modeling work (Piet et al, 2017), was interpreted to indicate that the FOF was not involved in integrating new evidence, but was instead involved in reading out the sign of the accumulation variable in service of reporting a choice via a motor movement.

In this work, we did not test whether the FOF played a role in evidence accumulation in a dynamic environment. Instead, we focused on the FOF's role in reading out the provisional decision in a dynamic environment where the provisional decision may change throughout the trial. Next, we wanted to strengthen the findings of Hanks et al (2015) by testing whether FOF units responded as predicted to changes of mind. This result would also provide validation of the evidence discounting behavioral model developed in Piet et al (2018), showing correspondence between model-predicted changes of mind and neural activity. When using model predicted changes of mind to align neural activity, we see changes in the firing rates that coincide with model-predicted changes in the provisional decision. The key finding of our study is not whether this encoding is reflecting a motor preparation signal versus the dynamic evidence accumulation, but rather that it changes at the specific time points predicted by our behavioral model. Testing changes in the provisional decision was not possible in the Hanks et

al (2015) study where the stationary environment minimized large fluctuations in the provisional choice within a trial.

We have changed the texts in several places to clarify these ideas to the reader. We will refer to each of these in our responses to the reviewer's specific points below.

Specific points

1. My first concern is that the title needs to be more precise with species, brain region and testing conditions. Rather than stating 'changes of mind' explicitly, which in my opinion was overly eye-catching than accurate, it would be helpful for the reader to know that the task has a dynamic nature which induces possible 'mind changes' as inferred by the experimenters, because there was a lack of direct behavioral evidence showing that rats indeed changed their "mind". 'Provisional or intended decision' might be more appropriate.

We have changed the title to include the relevant brain region and to indicate that we inferred changes of mind via a model. The new title is:

Stable choice coding in FOF across model-predicted changes of mind

2. It is unclear whether FOF is playing a different role in the current version of task comparing to the previous stationary accumulation task, and also comparing to mouse ALM in the delayed response task. In the current manuscript, the authors stated that FOF neurons encoded evidence, and tracked the evidence changes throughout the dynamic accumulation process. However, this appear to be at odds with the conclusion reached in the Hanks et al, 2015 study, where it was stated that "The more categorical encoding found in the FOF suggests that, contrary to current views, this brain region may not be involved in the graded evidence accumulation process itself, but instead may be more involved in the conversion to a categorical choice." This is basically motor preparation. This view was further supported by FOF inactivation experiments in Hanks et al, 2015, showing that only peri-choice inactivation led to a significant ipsilateral bias, while inactivation during the early accumulation period produced no significant effect. In my view, the notion of encoding evidence accumulation should be distinguished from the notion of encoding movement planing.

The reviewer raises an important concern about our presentation of the findings and their interpretation in light of the work of Hanks et al (2015). As discussed above, Hanks et al (2015) suggested that FOF did not integrate new evidence, but rather encoded the animal's provisional decision, providing a readout of the accumulation process. This signal may be best described as a motor preparation signal, or may be a motor-independent categorical signal. Our study builds on Hanks et al (2015) in two key ways that refine our understanding of the provisional decision read out. First, it was not clear before our study whether FOF units would still encode the provisional decision throughout the entire trial in a dynamic environment. Second, we find that fluctuations in neural activity correspond with predicted changes of mind from a model fit only to behavior, using an analysis that was not possible in a stationary environment.

We have modified the text to clarify how we interpret our results and how the work builds on that of Hanks et al (2015).

The modified abstract now reads (line 14):

How do we prepare to act in a constantly-changing environment? In the rat, provisional choice is thought to be represented in Frontal Orienting Fields (FOF), but this has only been tested in static environments where provisional and final decisions are not easily dissociated. Here, we characterize the representation of accumulated evidence in the FOF of rats performing a recently developed dynamic evidence accumulation task, which induces changes in the provisional decision. We find that FOF encodes evidence throughout decision formation with a temporal gain modulation that rises until the period when the animal may need to act. We find that reversals in FOF firing rates can be accounted for by the changes of mind predicted by a model of the decision process fit only to the behavioral data. Our results suggest that the FOF represents provisional decisions even in dynamic, uncertain environments, allowing for rapid motor execution when it is time to act.

The first 2 paragraphs of the modified introduction now read (line 26):

When making decisions, animals must weigh and combine the available evidence in favor of each alternative. With each new observation, evidence about the underlying state of the environment gradually accumulates until the animal is ready to act. This accumulation model successfully describes a wide array of decisions (Gold, 2007; Krajbich, 2015; Ratcliff, 1978). Neural correlates of this accumulation process are also present across many brain regions in animals performing perceptual categorization tasks (Gold, 2007; Brody, 2016). Not all brain regions with neural correlates of evidence accumulation play the same role in the decision making process (Brody, 2016; Siegel, 2015; Katz, 2016). For example, regions important for accumulation may represent evidence in a continuous, graded fashion. On the other hand, regions important for reading out choice and preparing motor movements may have more categorical representations of the accumulated evidence.

Hanks et al (2015) characterized the neural representation of accumulating evidence in rats performing accumulation of trains of auditory click evidence. In the task, two streams of randomly-timed auditory clicks were emitted from either side of a fixation location and rats were trained to orient toward the side that played a greater number of clicks. Presenting the evidence as discrete pulses provided additional power to estimate the evolution of each subject's latent accumulated evidence variable on individual trials (Brunton, 2013), increasing the resolution for estimating neural encoding of this variable across brain regions (Hanks, 2015; Scott, 2017; Yartsev, 2018). Experimenters recorded from the posterior parietal cortex (PPC), and the frontal orienting fields (FOF), a frontal cortical structure implicated in short term memory and preparation of orienting movements (Kopec, 2015; Erlich, 2011; Ebbesen, 2018). They found that FOF neurons encoded the instantaneous accumulated evidence with sigmoidal tuning curves that remained stable during accumulation (Hanks, 2015). These representations were more categorical than representations found in PPC, providing a readout of the animal's provisional decision—the choice favored by the evidence presented so far—throughout accumulation (Hanks, 2015; Yartsev, 2018). While this study could not differentiate between

evidence representations resulting from a role in motor preparation and motor-independent evidence representations, temporally-precise perturbations of the signals in FOF only impaired the animal's choice when they overlapped with the final time points of accumulation and not when they occurred early in the evidence period. These results, along with a two-node model of the FOF (Piet, 2017), suggested that the FOF is not involved in the accumulation of new pieces of evidence, but provides a critical readout of the animal's provisional decision when it is time to act.

The final paragraph of the introduction now reads (line 66):

Recent work has shown that rats and humans can learn to adopt the optimal discounting rate in a dynamic environment (Glaze et al, 2015; Piet et al, 2018). However, it is unknown whether the neural correlates of evidence accumulation observed during putatively non-leaky integration in stationary environments are preserved in animals performing putatively leaky integration in dynamic environments. Here, we recorded from FOF in rats during a dynamic accumulation of evidence task. We tested whether the stable code observed in the stationary environment persisted in the dynamic environment by applying and extending a method developed to characterize neural tuning to accumulated evidence (Hanks et al, 2015). The evolution of the latent accumulation variable was estimated using a behavioral model fit to the animal's choice data (Piet et al, 2018). In FOF, tuning to this accumulation variable was described by a single sigmoidal tuning curve multiplied by a time varying gain modulation, which increased with time early in the trial and stabilized at the time of the earliest possible go cue. We reasoned that if FOF neurons track the accumulated evidence throughout the entire accumulation period, firing rates should respond rapidly to changes in the provisional decision. Using the behavioral model to predict the precise timing of the animal's changes of mind, we found that FOF neurons respond within 100ms, reflecting the new provisional decision in their activity. By recomputing the evidence tuning curves aligned to these change of mind events, we confirmed that FOF neurons encode evidence with a single tuning curve before and after changes of mind. These results suggest that FOF maintains a stable readout of the decision provisionally favored by the accumulated evidence despite dynamic uncertainty in the environment and the upcoming choice. Maintaining a stable representation of the provisional decision may help ensure that the animal is ready when it is time to act. Our study opens up the opportunity for future work on the neural circuit level understanding of how animals integrate and decide in a volatile environment.

Although phenomenologically they are often difficult to separate, accurate perturbation experiment could provide important clues. Unfortunately, such manipulation experiment testing the causal role of FOF is missing in this manuscript.

We agree with the reviewer that perturbation experiments could provide a more detailed picture of the exact role played by the FOF during this task. For example, we could ask whether the FOF is only required at the last time points of evidence accumulation, like the Hanks study found in a stationary environment. However, as in Hanks et al (2015) perturbation experiments can not tell us whether an evidence signal is encoding a motor-independent representation of the evidence. For this, we would also need to modify the behavioral paradigm. While FOF

perturbations would clarify when the FOF is required for accurate behavior, they are unfortunately out of the scope of the present work.

In addition, in Hanks et al, 2015, comparing the information coding in PPC and FOF suggested that PPC is more likely to encode accumulated evidence. It would be helpful to also examine PPC activity in this dynamic accumulation task, and compare the activity between these two regions.

We agree with the reviewer that it would be interesting to determine whether the relationship between FOF (more categorical) and PPC (more graded) holds in the dynamic environment. However, PPC recordings are out of the scope of the present work.

3. Although this dynamic accumulation task is interesting in the sense of more closely simulating a volatile environment, this task also makes it more difficult to dissociate sensory evidence accumulation and motor planning. In this task, the choices were much more heavily dependent on the near-to-decision sensory information. According to Figure 1F, basically only the sensory information from the last 200 ms matters. This makes it difficult to distinguish whether a brain region is involved in sensory evidence accumulation or motor preparation, particularly for a region as FOF. The authors should explicitly discuss this and make more cautious interpretation of the information encoded in FOF neurons. Otherwise, the authors should provide additional analysis to distinguish whether FOF neurons were encoding movement signals or accumulated evidence, e.g., by looking at the FOF selectivity in error trials.

As discussed above, our interpretation of the results does not hinge on whether the FOF is involved in the accumulation process. We have attempted to make this clearer in the text in response to this reviewer's previous concerns and hope this helps resolve this issue.

Regarding the specific point about the sensory information from the last 200 ms being the most important, we would like to note that because the trials can end at any time (after the first 500ms), the rat is constantly incorporating new evidence and throwing out the old evidence. This is what allows us to predict the timing of changes of mind within a given trial and analyze firing rates around those time points.

To provide additional information about responses on error trials, we have added a supplementary figure (Fig. S6) containing the population average responses for correct trials and error trials conditioned on the cells' side-preferences. Consistent with a role for the FOF in dynamic representation of provisional decisions/motor preparation, the right panel in Fig. S6 shows that, immediately prior to movement onset, and regardless of how many "changes of mind" there were in a given trial, average firing rates in FOF are largely determined by which side port the animal has chosen to move to (i.e., corresponding motor act) and not the side favored by the experimenter-presented evidence.

We now discuss this in more detail when introducing the PSTHs in Figure 2 (line 166):

Many individual cells had trial-averaged firing rates that diverged throughout the trial, reaching a final value corresponding to the animal's choice. These cells had more intermediate average values throughout error trials, but eventually diverge according to the animal's choice, suggesting that firing rates reflect the animal's internal representation of the evidence or the motor plan.

Note that all the analysis in Main Text Figures 3, 4 and 5 include both correct and error trials--our behavioral model flexibly incorporates both, allowing us to examine neural activity based on the provisional decision and not the sensory evidence.

We now point this out explicitly when explaining tuning curve method (line 197):

Grouping firing rates according to the predicted accumulation values at each timepoint, allows us to more informatively combine information across trials with different hidden state change timing, final choice, and trial outcome (see Fig. S6 for population average of correct and error trials).

REVIEWER COMMENTS

Reviewer #1 (Remarks to the Author):

The authors have answered all my concerns and I congratulate them on a substantially clarified and improved paper. I make only one suggestion on exposition of the backward model, along with several minor typos and corrections.

Suggestion:

I very much appreciate the authors' work in expanding and clarifying their exposition of the model, particularly the posterior calculation, and I now better understand something that confused me when initially reading the paper: Properly speaking, as I noted previously, a_N is not distributed according to the uniform distribution, since that cannot be normalized over the interval $[B, \infty)$. Rather, what the authors mean is $p(a_N | a_N \in [B, \infty))$, which is the integral $\int_B^{\infty} da_N p(a_N | a)$. But even this, I suggest, is less intuitive for readers than simply thinking in terms of proportionality to the forward likelihood, as we would with hidden Markov Models or Kalman filtering. That is, $b(a) \propto p(a_N | a) = \int_B^{\infty} da_N p(a_N | a)$. That is, the past and the future are conditionally independent given the present, as the authors note, and their numerical calculation, despite operating backward, is just a means of calculating the forward likelihood from the present into the future. It is also easier to see in this formulation that one is just discretizing the integral over a_N above. I don't know how many readers may share my confusion, but this link was less clear to me on my first reading.

Minor:

- line 114: perhaps σ^2_0 to match a_0 ?
- Fig 3:
 - Caption for (C): You mean tuning averaged over time, right? Average X over time sounds like something is being plotted as a function of time.
 - There is no multiplication symbol between the last two panels of D (as in E)
 - line 246: "it's" should be "its"
 - Fig 4A, second panel: It took me a lot of staring to figure out that the shading indicated the probability density in the vertical direction. If these are Gaussian, why not just plot 1 and 2-standard deviation curves (or just one of these) along with the mean, since that gives the same information?
 - line 265: This is the expected value of a under the posterior, correct? In which case it's $E_{p(a)}[a]$?
 - line 434: Might help to indicate in text that $S(t)$ is the current state.
- ll. 448, 452, 457, etc.: I think the usual convention is to indicate variance in the normal, not standard deviation.

Reviewer #2 (Remarks to the Author):

The authors have addressed my concerns.

Reviewer #3 (Remarks to the Author):

1. In this revision, the authors have provided extensive textual adjustment and explanations that largely addressed my technical concerns. But my concern regarding the limited scope of scientific advancement in this study remains. The results represent a relatively small increment following the previous studies from the same lab. The main finding of this study that FOF activity was correlated with movement plan, even though the movement plans changes stochastically with stimulus state changes, is well expected from previous studies.^[1] It would be more interesting to examine whether and how a transient perturbation of the provisional activity would bias the animals' final choice or motor planning under dynamic environment. However, this type of experiment was considered by the authors as being out of the scope of this study.

2. Another concern that I had about the term “changes of mind” in the title and in the main text of this manuscript also remains. “Changes of mind” is not an accurate term for describing the experimental observations in this study. “Mind” is a vague term and difficult to define, and can be misleading for general readers, leading to the potential misconception that FOF neurons are encoding “mind” rather than motor preparation. In the behavioral experiments, one can only observe the animals’ choice, motor actions, or estimate the movement plans, but cannot observe or measure the “mind”. For this reason, I suggest the authors use “changes of movement plan” or similar terms instead of using “changes of mind”. It is not proper to lead the general audience to believe that that FOF neurons encode “mind”.

Response to Reviewers

Black, reviewer comments

Blue, our response

Purple, changes to manuscript text.

REVIEWER COMMENTS

Reviewer #1 (Remarks to the Author):

The authors have answered all my concerns and I congratulate them on a substantially clarified and improved paper. I make only one suggestion on exposition of the backward model, along with several minor typos and corrections.

Thank you for the positive evaluation of our work.

Suggestion:

I very much appreciate the authors' work in expanding and clarifying their exposition of the model, particularly the posterior calculation, and I now better understand something that confused me when initially reading the paper: Properly speaking, as I noted previously, a_N is not distributed according to the uniform distribution, since that cannot be normalized over the interval $[B, \infty)$. Rather, what the authors mean is $p(a|a_N \in [B, \infty))$, which is the integral $\int_B^{\infty} da_N p(a|a_N)$. But even this, I suggest, is less intuitive for readers than simply thinking in terms of proportionality to the forward likelihood, as we would with hidden Markov Models or Kalman filtering. That is, $b(a) \propto p(a_N \in [B, \infty)|a) = \int_B^{\infty} da_N p(a_N|a)$. That is, the past and the future are conditionally independent given the present, as the authors note, and their numerical calculation, despite operating backward, is just a means of calculating the forward likelihood from the present into the future. It is also easier to see in this formulation that one is just discretizing the integral over a_N above. I don't know how many readers may share my confusion, but this link was less clear to me on my first reading.

We again thank the reviewer for the careful reading of our notation. The reviewer is correct that the uniform distribution can't be normalized over the interval $[B, \infty)$, so we have clarified our notation. We changed the text in the results section when the posterior distribution is introduced (line 517):

$b(a)$ is the backward distribution, which assumes the final accumulation value is on the side of the decision boundary B corresponding to the animal's choice y :

And the cases in equation 19 are now written as

$$\begin{aligned} P(a|t, \delta_R, \delta_L, \theta, a_N \geq B), & \quad \text{if } y = 1 \\ P(a|t, \delta_R, \delta_L, \theta, a_N \leq B), & \quad \text{if } y = -1. \end{aligned}$$

The text was also modified at line 834:

However, when we compute the backward model, we begin at the final time point, where the final accumulation value is constrained to be on the side of the decision boundary corresponding to the animal's choice:

$$\begin{aligned} a_N &\geq B, & \text{if } y = 1 \\ a_N &\leq B, & \text{if } y = -1. \end{aligned}$$

We appreciate the reviewer's suggestion for making the presentation of the backward distribution more intuitive. However, we would prefer to keep the presentation as is as we find it more readily comprehensible (though, our opinion on this is admittedly a bit biased!). Additionally, the current presentation reflects our algorithmic computation of the posterior distribution.

Minor:

- line 114: perhaps σ_0^2 to match a_0 ?

The reviewer is correct that using σ_0^2 for the initial variance (instead of σ_i^2) would be internally consistent with our use of a_0 . However, previous papers using this model have used σ_i^2 for the initial variance, and for that reason we prefer to keep this notation.

- Fig 3:

- Caption for (C): You mean tuning averaged over time, right? Average X over time sounds like something is being plotted as a function of time.

Yes, we mean the tuning averaged over time. We have modified the caption to read:

(C) Tuning curve averaged over time. Points indicate mean (\pm s.e.m.) across time of the change in firing rate relative to temporal average as a function of accumulated evidence value a .

- There is no multiplication symbol between the last two panels of D (as in E)

We aren't sure what happened here. In our PDF there is a visible multiplication symbol between the last two panels of D (and E). We will ensure the final proofs include this symbol.

- line 246: "it's" should be "its"

Yes, thank you.

- Fig 4A, second panel: It took me a lot of staring to figure out that the shading indicated the probability density in the vertical direction. If these are Gaussian, why not just plot 1 and

2-standard deviation curves (or just one of these) along with the mean, since that gives the same information?

We thank the reviewer for this suggestion. However, the posterior distribution is not gaussian distributed, but rather a mixture of gaussians. To help clarify that $p(a)$ is a distribution at each time point, we have amended the caption to indicate the dependence on time.

We compute the posterior distribution of accumulated evidence given the choice at each time point, $p(a)$.

- line 265: This is the expected value of a under the posterior, correct? In which case it's $E_{p(a)}[a]$?

Yes, this is the expected value of a under the posterior. This notation would be more accurate, but it seems that the text description is adequate and the expression does not appear again. Therefore, we simply removed it. That sentence now reads (line 272).

For this analysis, model state changes were selected at time points when a 100ms running average of the posterior mean crossed the decision boundary B .

- line 434: Might help to indicate in text that $S(t)$ is the current state.

Thank you for a careful reading of our notation and methods. We amended the text to introduce $S(t)$ (line 442):

Then, the expected click difference rate given the current state of the environment, $E[d(t) | S(t)]$, was subtracted from d at each timepoint on each trial. Here, $S(t)$ is the current environmental state.

- ll. 448, 452, 457, etc.: I think the usual convention is to indicate variance in the normal, not standard deviation.

To our knowledge, both notations (standard deviation and variance) are used. We switched our notation to use variance since that appears to be more common.

Reviewer #2 (Remarks to the Author):

The authors have addressed my concerns.

Thank you.

Reviewer #3 (Remarks to the Author):

[A] concern that I had about the term “changes of mind” in the title and in the main text of this manuscript also remains. “Changes of mind” is not an accurate term for describing the experimental observations in this study. “Mind” is a vague term and difficult to define, and can be misleading for general readers, leading to the potential misconception that FOF neurons are encoding “mind” rather than motor preparation. In the behavioral experiments, one can only observe the animals’ choice, motor actions, or estimate the movement plans, but cannot observe or measure the “mind”. For this reason, I suggest the authors use “changes of movement plan” or similar terms instead of using “changes of mind”. It is not proper to lead the general audience to believe that that FOF neurons encode “mind”.

We agree with the reviewer that the phrase “changes of mind,” in its use of the word “mind,” can be somewhat vague. However, it seems that “change of mind” has become standard terminology in the literature that addresses the phenomena that our paper is concerned with. We want to make potential readers aware that our paper is part of this literature, so we very much would like to keep the term in the title and the manuscript. (Below we’ve included a list of some of the papers that, to our knowledge, use the term in the very same sense that we do. As you can see, it is not a short list! With multiple high profile papers in it.)

Nevertheless, we do appreciate the reviewer’s point. We have tried to address it by clarifying from the start, when the term is first used, that (consistent with its use in this literature) “change of mind” should be interpreted simply as shorthand for “change in the provisional decision”, and that we do not claim that the FOF encodes some abstract notion of “mind”. Further, where appropriate, we have replaced the phrase “changes of mind” with “model state change” to emphasize that in our study, changes in the provisional decision are inferred when the sign of our behavioral model’s latent state variable changes. This has greatly reduced the occurrence of the word “mind” in the text.

The third sentence of the abstract has been modified to clarify that “change of mind” is a shorthand for “changes in the provisional decision.” It now reads (line 17):

Here, we characterize the representation of accumulated evidence in the FOF of rats performing a recently developed dynamic evidence accumulation task, which induces changes in the provisional decision, referred to as “changes of mind”.

The final paragraph of the introduction, where the phrase “change of mind” is first used in the main text, now reads (line 78):

We reasoned that if FOF neurons track the accumulated evidence throughout the entire accumulation period, firing rates should respond rapidly to changes in the provisional decision, which in the literature are referred to for short as “changes of mind.” Such “changes of mind” have been studied in stationary environments when movement trajectories initiated toward one target are subsequently revised, possibly due to continued processing of the stimulus after initial decision commitment (Resulaj et al., 2009, van den

Berg et al., 2016). They may also arise from noisy fluctuations in decision-related neural activity (Atiya et al., 2019) and their timing may be inferred through neural decoding (Kiani et al., 2014, Peixoto et al., 2021). (For clarity, we emphasize that we do not claim to test whether the FOF encodes an abstract notion of “mind”, but much more simply that changes in the provisional decision can be read out from FOF activity.) We used a behavioral approach to predict the precise timing of changes of mind using the latent state of the behavioral model fit to each rat's choice data. We found that FOF neurons responded to these model state change events within 100ms, reflecting the new provisional decision in their activity. Recomputing the evidence tuning curves aligned to model state changes, we confirmed that FOF neurons encode evidence with a single tuning curve before and after changes of mind.

Finally, the title has been modified to “Stable choice coding in rat FOF across model - predicted changes of mind” to clarify our change of mind events are predicted from a model and come from behavioral measurements in rodents.

Here is a summary of neuroscience papers that use the phrase “changes of mind”. This is a non-exhaustive list that highlights similar studies. We include a quote from each paper highlighting their usage of the phrase “change of mind”. Emphasis ours.

Loffler et al. (2021). A Hierarchical Attractor Network Model of perceptual versus intentional decision updates. *Nature Communications*.

- “Movement trajectories revealed whether and when participants changed their mind about the dot-motion direction, or additionally **changed their mind** about which colour to choose.”

Rollwage et al. (2020). Confidence drives a neural confirmation bias. *Nature Communications*.

- “confidence shapes a selective neural gating for choice-consistent information, reducing the likelihood of **changes of mind** on the basis of new information.”

Murphy et al., (2021). Adaptive circuit dynamics across human cortex during evidence accumulation in changing environments. *Nature Neuroscience*.

- “due to possible **changes of mind**, decision states earlier in the trial could differ substantially from the choice-determining state toward the end of the trial”

Peixoto et al., (2021). Decoding and perturbing decision states in real time. *Nature*.

- “When making a categorical decision about a noisy stimulus, subjects commonly fluctuate between levels of commitment to a choice before reporting a decision. In some instances, the fluctuations are sufficiently strong to lead to a **change of mind**”

Onagawa et al., (2021) Sensorimotor strategy selection under time constraints in the presence of two motor targets with different values. *Scientific Reports*.

- “the longer the time constraint, the higher the frequency of the intermediate behavior (to initiate movement toward the intermediate direction of two targets) or the **change-of-mind** behavior (to change the aiming target during movement)”

Lee et al., (2021) Trading mental effort for confidence in the metacognitive control of value-based decision-making. *eLife*.

- “**change of mind** according to two criteria: (i) the choice is incongruent with the prior preference inferred from the

pre-choice value ratings, and (ii) the choice is congruent with the posterior preference inferred from post-choice value ratings”

Turner *et al.*, (2021) Perceptual **change-of-mind** decisions are sensitive to absolute evidence magnitude. *Cogn. Psychology*.

- “it is unknown whether **change-of-mind** decisions are sensitive to absolute evidence. Here, we investigated this question across two experiments. In each experiment participants indicated which of two flickering greyscale squares was brightest. Following an initial decision, the stimuli remained on screen for a brief period and participants could change their response.”

Cos *et al.*, (2021) **Changes of Mind** after movement onset depend on the state of the motor system. *eNeuro*.

- “we applied mechanical perturbations to the arm during movement to study under which conditions such perturbations produce **changes of mind**. Our results show that participants initially selected the direction of movement towards the highest reward region, and **changed their mind** most frequently when the two choices offered the same reward”

Loffler *et al.*, (2020) Decoding Changes of Mind in Voluntary Action-Dynamics of Intentional Choice Representations. *Cerebral Cortex*.

- “previous studies investigating **CoM** have predominantly focused on perceptual decisions, which are driven by external evidence (e.g., Resulaj *et al.* 2009). Conversely, despite its utmost importance for everyday life, little is known about the neurocognitive mechanisms underlying **CoM** in voluntary action.”

Atiya *et al.*, (2020) **Changes-of-mind** in the absence of new post-decision evidence. *PLOS Computational Biology*.

- In a motion discrimination task, we demonstrate that **changes-of-mind** can occur even in the absence of additional evidence after the initial decision

Atiya *et al.*, (2019). A neural circuit model of decision uncertainty and **change-of-mind**. *Nature Communications*.

- “**Changing one’s mind** has been attributed to processing new evidence that negates a previous judgement. More recent neurophysiological evidence has shown that some **changes-of-mind** occur as a result of an internal error-correction mechanism, suggesting decision uncertainty plays a role in inducing **changes-of-mind**”

Fleming *et al.*, (2018). Neural mediators of **changes of mind** about perceptual decisions. *Nature Neuroscience*.

- “This design allowed us to study the underpinnings of **changes of mind** by analyzing how new evidence impacts confidence bidirectionally, in a graded fashion, rather than only on a subset of trials on which discrete choice reversals are observed”

Rollwage *et al.*, (2018) Metacognitive Failure as a Feature of Those Holding Radical Beliefs. *Current Biology*.

- “This model accounts for a reduction in post-decisional processing in more radical participants by boosting confidence in chosen options, thereby making **changes of mind** less likely”

van den Berg *et al.*, (2016) A common mechanism underlies **changes of mind** about decisions and confidence. *eLife*.

- “we also observed revisions of the initial choice and/or confidence. These **changes of mind** were explained by a continuation of the mechanism that led to the initial choice.:

Kaufman *et al.*, (2015) Vacillation, indecision and hesitation in moment-by-moment decoding of monkey motor cortex. *eLife*.

- “we observed presumed ‘**changes of mind**’: the neural state initially reflected one choice before changing to reflect the final choice.”

Bronfman *et al.*, (2015) Decisions reduce sensitivity to subsequent information. *Proceedings of the Royal Society B*

- “final evaluations that were opposite to their preliminary decision on a substantial fraction of trials (M = 32%; s.d. = 8%; the minimal observed fraction was 16%); 72% of these ‘**changes of mind**’ occurred on trials in which the mean of the additional information was incongruent with the participants’ preliminary decision.”

Wei *et al.*, (2015) Confidence estimation as a stochastic process in a neurodynamical system of decision making. *J. Neurophysiology*.

- “Interestingly, the model predicts that **changes of mind** can occur in a mnemonic delay when confidence is low; the probability of **changes of mind** increases (decreases) with task difficulty in correct (error) trials”

Burk *et al.*, (2014) Motor effort alters **changes of mind** in sensorimotor decision making. *PLOS one*.

- “After committing to an action, a decision-maker can **change their mind** to revise the action.”

Kiani *et al.*, (2014) Dynamics of neural population responses in prefrontal cortex indicate **changes of mind** on single trials. *Current Biology*.

- “On individual trials, the decoded DV varied significantly over time and occasionally changed its sign, identifying a potential **CoM**”

Moher *et al.*, (2014) Perceptual decision processes flexibly adapt to avoid **change-of-mind** motor costs. *J. Vision*.

- Although each response box was always equidistant from the starting position, the physical distance between the two response options was varied. We found that when the boxes were far apart from one another, and thus **changes of mind** incurred greater redirection motor costs, **change-of-mind** frequency decreased while latency to initiate movement increased

Bollimunta *et al.*, (2012) Neural dynamics of choice: single-trial analysis of decision-related activity in parietal cortex. *J. Neuroscience*.

- “we identify individual decisions that can be described as a **change of mind**: the decision circuitry was transiently in a state associated with a different choice before transitioning into a state associated with the final choice”

Albantakis *et al.*, (2012). A multiple-choice task with **changes of mind**. *PLOS One*.

- “**Changes of mind** in the participants’ movement trajectories could be observed for two and for four choice alternatives.”

Resulaj *et al.*, (2009) **Changes of mind** in decision-making. *Nature*.

- “Although they received no additional information after initiating their movement, their hand trajectories betrayed a **change of mind** in some trials. We propose that noisy evidence is accumulated over time until it reaches a criterion level, or bound, which determines the initial decision, and that the brain exploits information that is in the processing pipeline when the initial decision is made to subsequently either reverse or reaffirm the initial decision.”

In this revision, the authors have provided extensive textual adjustment and explanations that largely addressed my technical concerns. But my concern regarding the limited scope of scientific advancement in this study remains. The results represent a relatively small increment following the previous studies from the same lab. The main finding of this study

that FOF activity was correlated with movement plan, even though the movement plans changes stochastically with stimulus state changes, is well expected from previous studies.

Respectfully, we disagree that our findings were well expected from previous studies. To our knowledge, cortical responses to evidence accumulation in a dynamic environment had not previously been studied in animal subjects. Our study could have found that in a dynamic environment the FOF does not begin representing the provisional decision/motor action until a final decision has been reached. Further, we would like to emphasize that responses in FOF are not just correlated to the stimulus state changes, but to the latent variable of our behavioral model. Notably, FOF responses are better aligned to the timing of model state changes than to the timing of changes in the generative environment (i.e., the stimulus state changes) (Fig. 4E). Finally, we arrive at these conclusions through advances in behavioral modeling and analysis of spike trains.

It would be more interesting to examine whether and how a transient perturbation of the provisional activity would bias the animals' final choice or motor planning under dynamic environment. However, this type of experiment was considered by the authors as being out of the scope of this study.

We agree with the reviewer that a perturbation study would be interesting and informative. However, we continue to feel such an experiment, which would be quite extensive, is out of scope for this study.

REVIEWER COMMENTS

Reviewer #1 (Remarks to the Author):

I appreciate the authors' revisions and defer on the matters of taste. All my substantive concerns were addressed in the previous version.

I also agree with the authors that, for good or ill, "change of mind" has been used in the literature on this sort of decision in non-human animals for some time. In my view, situating the term in its context, as their revised manuscript does, is sufficient.

Response to Reviewers

Black, reviewer comments

Blue, our response

Purple, changes to manuscript text.

REVIEWER COMMENTS

Reviewer #1 (Remarks to the Author):

I appreciate the authors' revisions and defer on the matters of taste. All my substantive concerns were addressed in the previous version.

I also agree with the authors that, for good or ill, "change of mind" has been used in the literature on this sort of decision in non-human animals for some time. In my view, situating the term in its context, as their revised manuscript does, is sufficient.

We thank the reviewer for detailed feedback that greatly improved the clarity of the manuscript and for the positive evaluation of our work.